# VGLUT2-expressing neurons in the vestibular nuclear complex mediate gravitational stress-induced hypothermia in mice

Chikara Abe[1,4 ✉], Yusuke Yamaoka[1,4], Yui Maejima[1], Tomoe Mikami[1], Shigefumi Yokota[2], Akihiro Yamanaka[3] & Hironobu Morita[1 ✉]

The vestibular system, which is essential for maintaining balance, contributes to the sympathetic response. Although this response is involved in hypergravity load-induced hypothermia in mice, the underlying mechanism remains unknown. This study showed that hypergravity (2g) decreased plasma catecholamines, which resulted in hypoactivity of the interscapular brown adipose tissue (iBAT). Hypothermia induced by 2g load was significantly suppressed by administration of beta-adrenergic receptor agonists, suggesting the involvement of decrease in iBAT activity through sympathoinhibition. Bilateral chemogenetic activation of vesicular glutamate transporter 2 (VGLUT2)-expressing neurons in the vestibular nuclear complex (VNC) induced hypothermia. The VGLUT2-expressing neurons contributed to 2g load-induced hypothermia, since their deletion suppressed hypothermia. Although activation of vesicular gamma-aminobutyric acid transporter-expressing neurons in the VNC induced slight hypothermia instead of hyperthermia, their deletion did not affect 2g load-induced hypothermia. Thus, we concluded that 2g load-induced hypothermia resulted from sympathoinhibition via the activation of VGLUT2-expressing neurons in the VNC.

[1] Department of Physiology, Gifu University Graduate School of Medicine, Gifu, Japan. [2] Department of Anatomy and Neuroscience, Shimane University School of Medicine, Izumo, Shimane, Japan. [3] Department of Neuroscience II, Research Institute of Environmental Medicine, Nagoya University, Nagoya, Japan. [4] These authors contributed equally: Chikara Abe, Yusuke Yamaoka. ✉email: chikara@gifu-u.ac.jp; zunzunmorita@gmail.com

The vestibular system is a known gravity sensory system in fish, reptiles, birds, and mammals[1]. This dynamic peripheral sensor, located in the inner ear, consists of two components, namely the semicircular canals and otolith organs, which detect angular and linear accelerations, respectively. The otolith organs also detect head position (static sensor) with respect to gravitational force. The signals from each sensor are transmitted to the brain, specifically to the vestibular nuclear complex (VNC), through the vestibular nerve[2]. The vestibular system contributes to both eye movement (i.e., the vestibulo-ocular reflex) and posture (i.e., the vestibulo-spinal reflex), and is important for the understanding of the body's dynamics and kinematics[3].

Gravitational change is detected by the peripheral vestibular organs. The stimulation of the peripheral vestibular organs is known to induce the sympathetic nervous response (vestibulo-sympathetic reflex)[4,5]. The delivery of electrical stimuli to the branches of the vestibular nerve elicits changes in sympathetic nerve activity including excitation, inhibition, or a combination of the two[4]. Hypergravity is one of the conditions that activates the peripheral vestibular organs. It seems that short-term exposure to hypergravity activates the sympathetic nervous system[6], while chronic exposure to hypergravity may induce sympathoinhibition. Fuller et al. were the first to demonstrate that chronic exposure to 2g environment-induced hypothermia was suppressed in het mice, which lack macular otoconia[7]. Monson et al. reported that exposure to 2.1g increased tail temperature, accompanied by a decrease in body temperature (BT) in rats, suggesting an sympathoinhibition-induced increase in tail blood flow[8]. Furthermore, they also showed that hypergravity decreased oxygen consumption, which resulted in suppression of heat production, with a possible involvement of sympathoinhibition-induced hypoactivation of the adrenergic receptors in the brown adipose tissues. However, there is no direct evidence that chronic stimulation of the peripheral vestibular organs induced by hypergravity decreases heat production via sympathoinhibition.

Although several studies have demonstrated the existence and functionality of the vestibulo-sympathetic reflex, relatively little is known about the specific connectivity of the neurons in the brain. Holstein et al. demonstrated that there is a direct projection from the caudal vestibular nuclei to the rostral ventrolateral medulla (RVLM), which is a cluster of presympathetic neurons[9]. We also reported that calcium/calmodulin-dependent protein kinase II (CAMK2)-expressing neurons in the VNC project to the RVLM. C1 neurons probably participate in the vestibulo-sympathetic reflex[10]. Paraventricular nucleus neurons also include presympathetic neurons. These neurons project to the preganglionic sympathetic neurons in the spinal cord and RVLM and are possibly involved in regulating sympathetic outflow and blood pressure[11,12]. The neurons in the hypothalamus seems to be influenced by the vestibular inputs[4]. Expression of c-fos in paraventricular nucleus induced by hypergravity was suppressed in het mice[13] and rats with vestibular lesions (VL)[14], suggesting that the presympathetic neurons in the hypothalamus may be also involved in vestibulo-sympathetic reflex. Intense glutamate immunofluorescence in VNC neurons activated by sinusoidal galvanic vestibular stimulation (GVS) was observed in anatomical experiments. These neurons project to the presympathetic regions in the RVLM; however, their function is still unclear.

In the present study, we found that chronic stimulation of the peripheral vestibular organs induced by a chronic 2g load decreased plasma catecholamines, which resulted in hypothermia through the hypoactivation of interscapular brown adipose tissue (iBAT). We also elucidated the neurophysiological mechanism of 2g load-induced hypothermia through a chemogenetic approach. Vesicular glutamate transporter-2 (VGLUT2)-expressing neurons in the VNC have a crucial role in vestibular system-related thermoregulation.

## Results

**Hypothermia induced by 2g load is decreased plasma catecholamine levels.** Our experiments first focused on the afferent and efferent mechanisms underlying hypergravity-induced hypothermia, to identify the mechanism underlying vestibular system-related thermoregulation. Although previous studies on the afferent mechanism showed that otoconia deletion in global knockdown mice (mutation of NADPH oxidase 3) suppressed hypergravity-induced hypothermia[7], the effect of local disruption has not been described. Therefore, the BT and activity in the 2g environment of mice with VL[15] and sham-operated mice (Sham) were measured for 7 days. A significant decrease in BT was observed in Sham mice, and this response was significantly suppressed in VL mice on the first day of exposure to the 2g environment (Fig. 1a–d). The peripheral vestibular organs may also have a potential influence on the regulation in BT rhythms (Fig. 1a, b), which is consistent with prior findings[16]. We compared the BT response between on-axis and off-axis rotations to examine whether 2g-induced hypothermia was mediated by the otolith organs or semicircular canals (Supplementary Fig. 1). Although on-axis rotation also decreased BT, the effect was significantly smaller than that for off-axis rotation, suggesting that the otolith organs mainly contribute to 2g-induced hypothermia (Fig. 1e and Supplementary Fig. 1) as shown previously by Fuller et al[7]. Furthermore, we examined the effect of gravitational slope from 1g to 2g on hypothermia. Although 2g conditions were maintained for 48 h, they were created within 10 min, 6 h, 24 h, and 48 h. Hypothermia induced by exposure to 2g in 48 h was significantly attenuated compared with that in 10 min, 6 h, and 24 h (Fig. 1f, g). Although body mass decreased significantly decreased after exposure to 2g for 48 h, food intake was maintained (Supplementary Fig. 2), suggesting that the degree of motion sickness (estimated by decrease in BT and food intake) may be suppressed by the slow increase in gravity from 1g to 2g.

Hypothermia may result from an increase in heat loss or a decrease in heat production[17]. Thermography of mice under 2g revealed that both processes (heat loss from the tail and decreased heat production from the iBAT) contributed to hypothermia (Fig. 2a). Since heat production through the iBAT and blood flow in the tail are affected by the sympathetic nervous system[18,19], we measured BT through administration of the ganglionic blocker, hexamethonium (Supplementary Table 1). Hexamethonium produced similar responses (Fig. 2a, b, Supplementary Fig. 3), suggesting that a reduction in sympathetic nerve activity might be involved in 2g-induced hypothermia. A previous study showed that tailless rats did not show attenuation of hypergravity load-induced hypothermia[20]. In contrast, the temperature of the iBAT, which was measured by an implantable device, was significantly decreased in Sham but not in VL mice (Fig. 2c). Since their heart rates also decreased during exposure to the 2g environment (Supplementary Fig. 4), we hypothesized that decrease in either sympathetic nerve activity, circulatory catecholamines, or both might have occurred during 2g exposure. We performed blood sampling after anesthetizing the mice in 2g, to prevent the formation of artifacts induced by the cessation of the 2g load and restraint. The concentration of noradrenaline and adrenaline during exposure to 2g decreased significantly decreased in the Sham mice but not in VL mice (Fig. 2d). We used alpha and/or beta-adrenoreceptor agonists or alpha-2 adrenoreceptor antagonists to examine whether adrenergic agonists could reduce hypothermia induced by the 2g load (Supplementary Table 1). Since isoprenaline, adrenaline, and the beta-3 adrenoreceptor

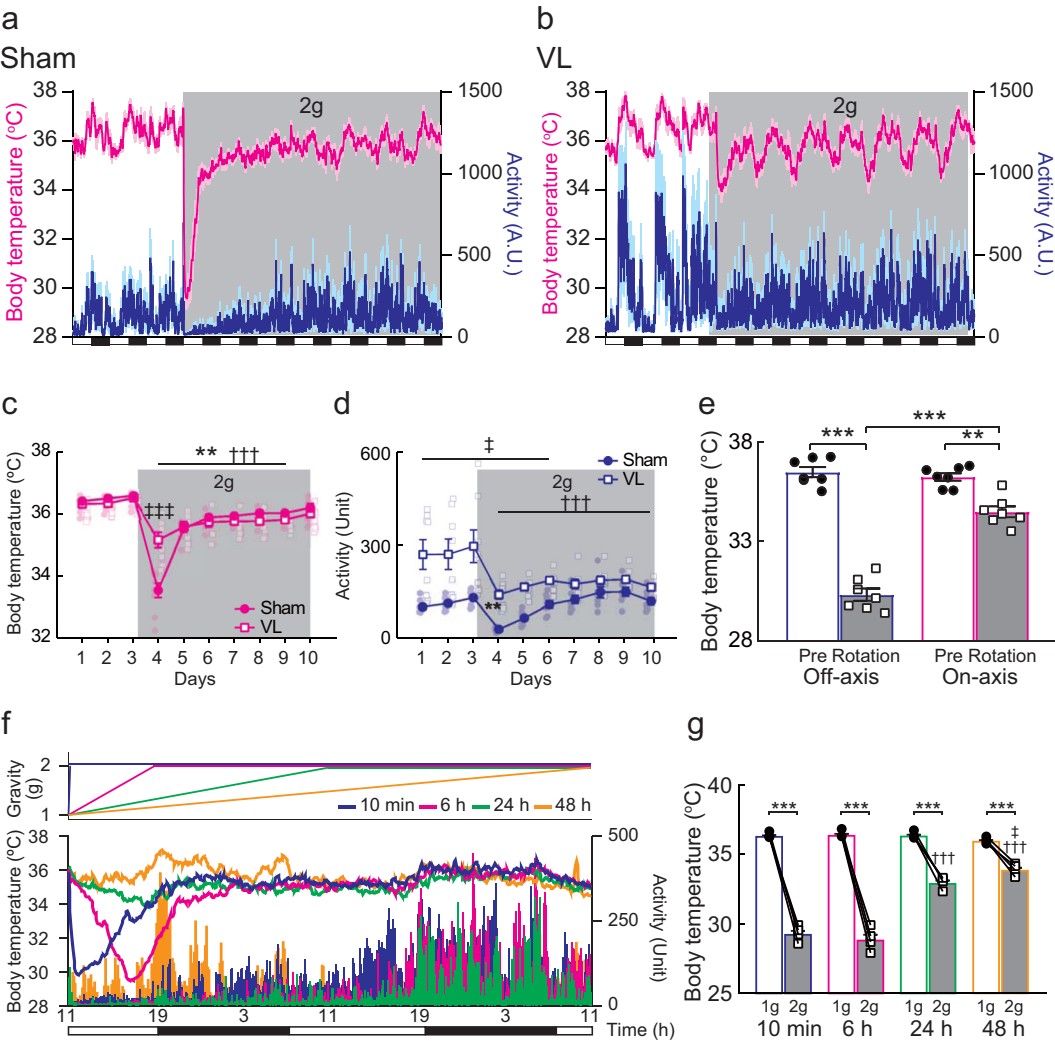

**Fig. 1 Effect of local disruption of otolith organs on hypergravity (2g)-induced hypothermia. a, b** Averaged data of the body temperature (BT, magenta) and activity (blue) prior to and during the 2g load (n = 8) in mice with a vestibular lesion (VL) and sham-operated mice (Sham). The horizontal open and closed squares indicate the light and dark periods, respectively. **c, d** Summarized data of the changes in BT (**c**) and activity (**d**) induced by 2g load in the Sham (n = 8) and VL (n = 8) mice. Two-way ANOVA with either Bonferroni's or Tukey's post-hoc test. * or † vs. averaged value of pre-loading in the Sham or VL mice, respectively; ‡ Sham vs. VL. **e** Summarized data of the changes in BT during either on-axis or off-axis rotation (n = 7 for each). Two-way ANOVA with Bonferroni's multiple comparisons test. **f** Hypergravity (2g) was created within 10 min, 6 h, 24 h, and 48 h. The BT and activity were recorded for 48 h (from 11 am to 11 am). The graph with a line indicates the BT, while the bar graph depicts activity. The horizontal bar under the numbers (Time) represents the light-and-dark cycle. **g** Summarized data of BT induced by a slow increase in gravity from 1g to 2g in mice (n = 4). Two-way ANOVA with the Tukey's and Bonferroni's multiple comparisons tests. For all statistical analyses, single, double, or triple significant symbols indicate P < 0.05, P < 0.01, or P < 0.001, respectively. Detailed information on the statistical analyses is reported in Supplementary Table 2.

agonist, BRL-37344, suppressed the hypothermic response (Fig. 2e), inhibition of the sympathetic nervous system and/or decrease in catecholamine release from the adrenal medulla may be involved in 2g-induced hypothermia.

**Role of VGLUT2- and VGAT-expressing neurons in the VNC in balance function.** We investigated the possible central mechanism, which included the VNC, considering that it contains glutamatergic and GABAergic/glycinergic neurons (Supplementary Fig. 5)[21]. Optogenetic tools were used (using a revised procedure of our previous methods) to determine whether these neurons contributed to vestibular function (Supplementary Fig. 5)[22]. The viral vector, AAV–DIO–EF1α–ChR2–mCherry serotype 2 or AAV–EF1a–DIO–eArch3.0–eYFP serotype 2, was unilaterally injected to induce opsin expression in the VNC (Fig. 3a). Unilateral excitation induced body tilt to the ipsilateral side, while inhibition

had the opposite effect in VGLUT2-Cre mice (Fig. 3b, c, and Supplementary Movie 1). Excitation of vesicular gamma-aminobutyric acid transporter (VGAT)-expressing neurons present in the VNC produced a contralateral tilt and inhibition caused an ipsilateral tilt. These responses were contrary to those observed on photostimulation of VGLUT2-expressing neurons (Fig. 3b, d, and Supplementary Movie 1). Moreover, increasing the stimulation frequency of the VGLUT2 or VGAT-expressing neurons enhanced the angle of body tilt (Fig. 3e, f).

Chemogenetic tools were used to evaluate the effect of chronic stimulation of VGLUT2- and VGAT-expressing neurons present in the VNC on behavior to further investigate the phenomenon. The viral vector, AAV–CAG–FLEX–hm3D(Gq)–mCherry, was unilaterally injected in the VNC of VGLUT2-Cre and VGAT-Cre mice. The receptors for chemogenetic stimulation [i.e., hm3D (Gq)] were expressed in either VGLUT2- or VGAT-expressing

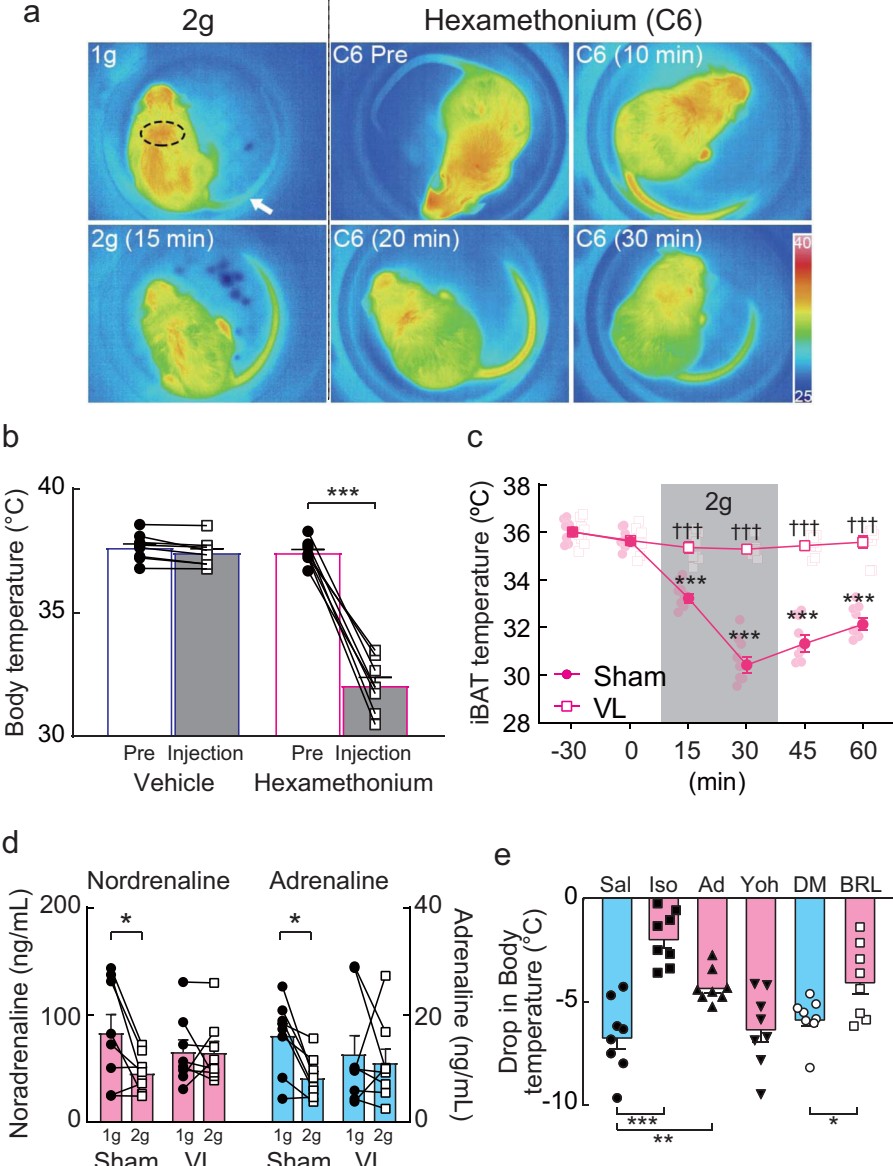

**Fig. 2 Efferent mechanism underlying hypergravity (2*g*)-induced hypothermia. a** Representative temperature response of the intrascapular brown adipose tissue (iBAT, dotted circle) and tail (arrow) following either 2*g* load or the administration of hexamethonium. **b** Summarized data of the changes in the body temperature (BT) of mice after administration of either hexamethonium or saline ($n = 8$, respectively). Two-way ANOVA with Bonferroni's multiple comparison test. **c** Summarized data of the changes in the iBAT temperature induced by 2*g* load in the Sham ($n = 8$) and vestibular lesion (VL) ($n = 8$) mice. Two-way ANOVA with either Bonferroni's or Tukey's post-hoc test. * vs. averaged value of the pre-loading; † vs Sham. **d** Summarized data of the changes in noradrenaline (magenta) and adrenaline (blue) induced by 2*g* load in the Sham and VL mice. Two-way ANOVA with Bonferroni's post-hoc test. * 1*g* vs. 2*g*. **e** The effect of autonomic drugs on 2*g* load-induced hypothermia. Sal, saline; Iso, isoprenaline; Ad, adrenaline; Yoh, Yohimbine; DM, dimethyl sulfoxide; BRL, BRL-37344. One-way ANOVA with either Bonferroni's post-hoc test or unpaired *t*-test. For all statistical analyses, single, double, or triple significant symbols indicate $P < 0.05$, $P < 0.01$, or $P < 0.001$, respectively. Detailed information on the statistical analyses is reported in Supplementary Table 2.

neurons located in the left VNC, while they were not observed in the corresponding regions of the C57BL/6J mouse (Supplementary Fig. 6). Clozapine N-Oxide (CNO)-induced activation of VGLUT2-expressing neurons led to a counterclockwise rotation along the edge of the cage and spinning behavior, although no difference was observed in the total distance of the open field compared to that in the control group (Fig. 4a, c, and Supplementary Movies 2 and 4). In contrast, activation of VGAT-expressing neurons located in the VNC was associated with freezing behavior and tottering with body tilted to the right side (contralateral to the activation site) (Fig. 4b, c, and

Supplementary Movies 3 and 4). Moreover, activation of VGLUT2- and VGAT-expressing neurons present in the VNC attenuated the rotarod skill 1 h after CNO administration, suggesting that unilateral activation of either VGLUT2- or VGAT-expressing neurons in the VNC disrupts vestibular-system-related motor coordination (Fig. 4d). This decrease in rotarod skill was recovered fully 24 h after CNO delivery. Therefore, VGLUT2- and VGAT-expressing neurons present in the VNC contribute to balance. The role of each neuron responsible for balance is opposite, even when they are in the same area in the VNC.

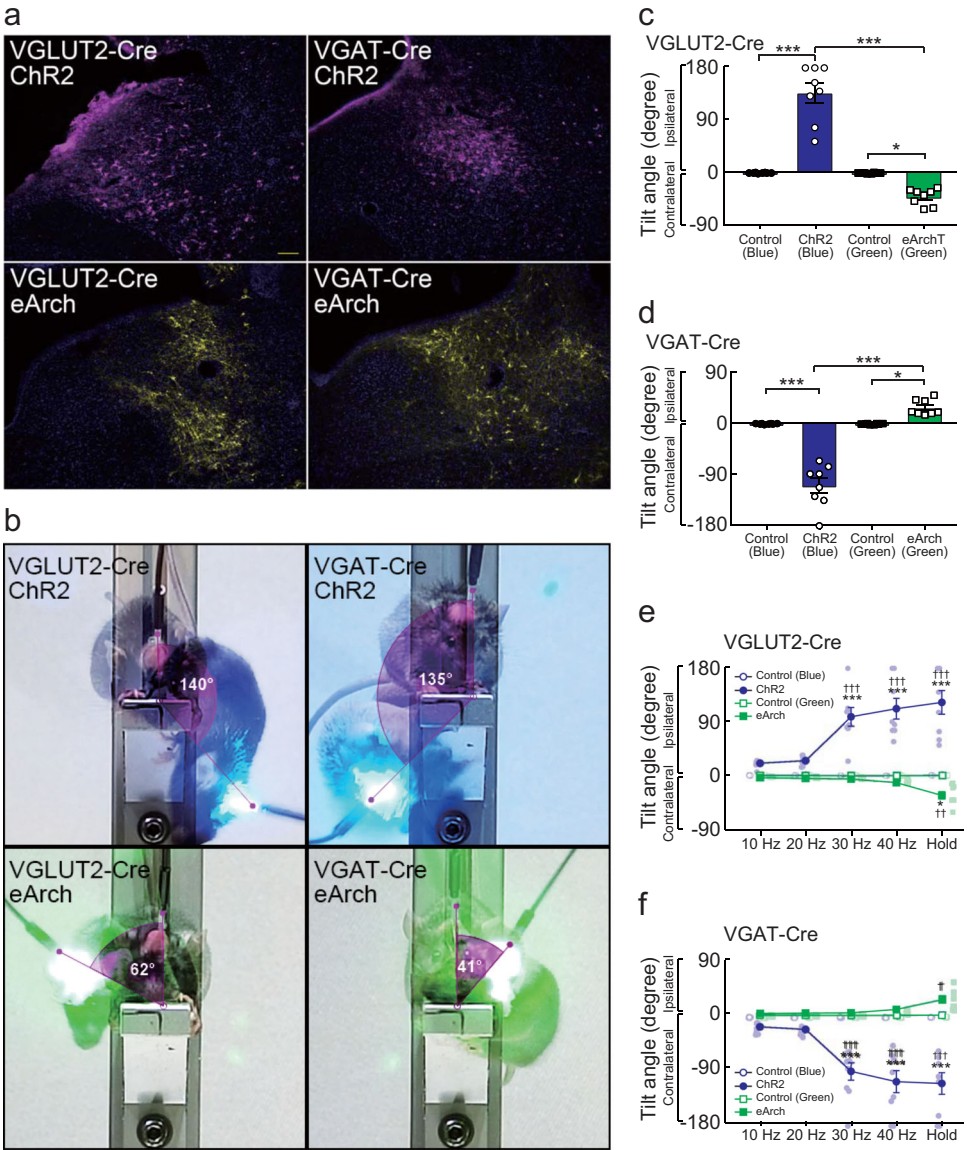

**Fig. 3 Manipulation of VGLUT2- and VGAT-expressing neurons located in the vestibular nuclear complex (VNC) using both the optogenetics tool.**
**a** Representative images of either ChR2 or eArch expression in VGLUT2- and VGAT-expressing neurons in the VNC. The scale bar is 100 μm.
**b** Photostimulation of either the ChR2 (blue) or the eArch (green) in VGLUT2- and VGAT-expressing neurons located in VNC-induced body tilt. The photostimulation was applied in the unilateral VNC. Please see the Supplementary Movie 1. **c, d** Summarized data of the changes in body tilt angle induced by photostimulation in either VGLUT2-Cre (**c**) or VGAT-Cre (**d**) mice. One-way ANOVA with Tukey's post-hoc test. **e, f** Summarized data of the body tilt angle induced by changes in the frequency of photostimulation in either VGLUT2-Cre (**e**) or VGAT-Cre (**f**) mice. Two-way ANOVA with Tukey's multiple comparisons test. * vs. 10 Hz and 20 Hz, † vs. Control. For all statistical analyses, single, double, or triple significant symbols indicate $P < 0.05$, $P < 0.01$, or $P < 0.001$, respectively. Detailed information on the statistical analyses is reported in Supplementary Table 2.

**Role of VGLUT2- and VGAT-expressing neurons in the VNC on thermoregulation.** The BT of the mice and activity during the activation of each subset of neurons were measured with chemogenetics to examine the role of VGLUT2- and VGAT-expressing neurons in the VNC on thermoregulation. The viral vector, AAV–CAG–FLEX–hm3D(Gq)–mCherry, was injected bilaterally in the VNC in the VGLUT2-Cre and VGAT-Cre mice. Bilateral activation of VGLUT2-expressing neurons located in the VNC significantly increased the distance traveled (Supplementary Fig. 7). However, contrary to the unilateral activation of VGLUT2-expressing neurons, bilateral stimulation induced a clockwise and a counterclockwise rotation along the edge of the cage (Supplementary Movie 5). In contrast, bilateral activation of VGAT-expressing neurons in VNC showed freezing behavior,

which was similar to the response to unilateral activation (Supplementary Fig. 7). Chemogenetic activation of VGLUT2- and VGAT-expressing neurons in the unilateral or bilateral VNC decreased the rotarod skill; this attenuation was fully restored in 24 h (Fig. 4d and Supplementary Fig. 7). A decrease in BT was observed following CNO administration in VGLUT2-Cre mice, considering the thermal response, although the activity was increased (Fig. 5a, c, d). Although this phenomenon was also observed during unilateral activation of VGLUT2-expressing neurons in the VNC, a smaller decrease in BT was observed (Supplementary Fig. 8). Unilateral activation of VGAT-expressing neurons located in the VNC increased BT, while unilateral inhibition decreased BT (Supplementary Fig. 8). On the other hand, bilateral activation resulted in a slight decrease in BT and

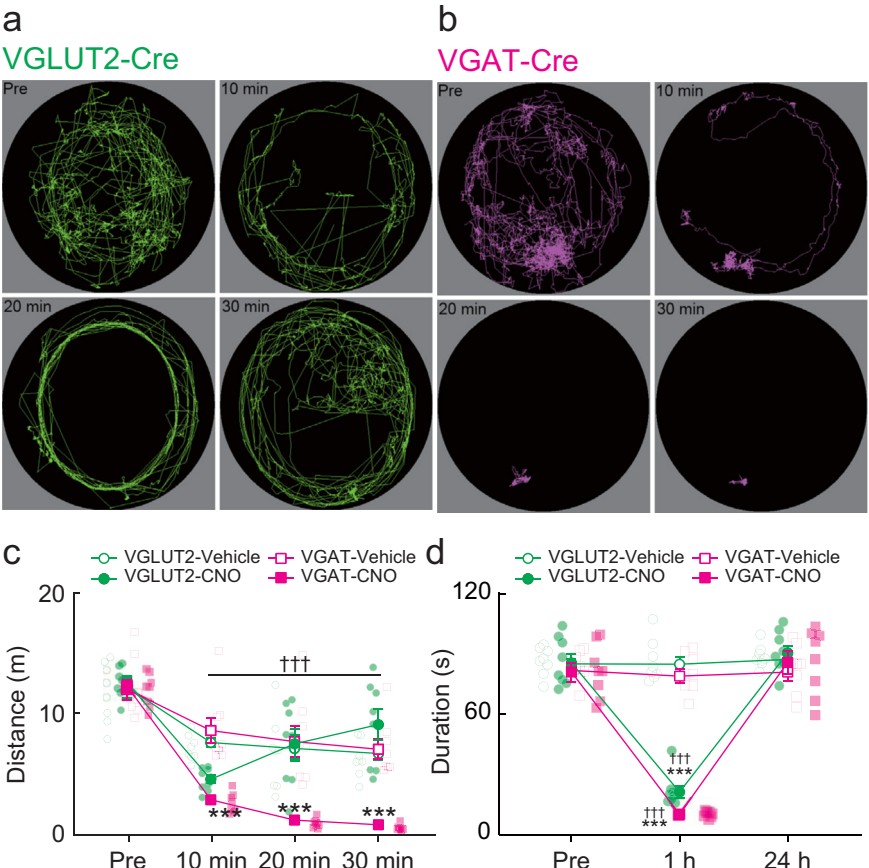

**Fig. 4 Manipulation of VGLUT2- and VGAT-expressing neurons located in the vestibular nuclear complex (VNC) using the chemogenetics tool.**
**a**, **b** Representative moving trace prior to and following clozapine N-oxide (CNO) injection in VGLUT2-Cre and a VGAT-Cre mouse. The receptors for chemogenetics (i.e., hm3D(Gq)) were expressed in either VGLUT2- or VGAT-expressing neurons located in the VNC. The chemogenetic stimulation was applied in the unilateral VNC. Please see the file in Supplementary Move 2 and 3. **c** Summarized data of the changes in the total moving distance for 10 min prior to and following injection of CNO ($n = 8$ in each group). Two-way ANOVA with Tukey's post-hoc test. * vs. VGAT-vehicle; † vs. pre-loading in all groups. **d** The duration of the rotarod experiment was measured prior to and following the administration of either clozapine N-oxide (CNO) or saline (vehicle). Each mouse underwent three sessions of the rotarod experiment, for which the averaged value was used. Two-way ANOVA with Tukey's multiple comparison tests. * vs. Pre, † vs. either VGLUT2-vehicle or VGAT-vehicle. For all statistical analyses, triple significant symbols indicate $P < 0.001$. Detailed information on the statistical analyses is reported in Supplementary Table 2.

activity (Fig. 5b, c), although the decline began 120 min after CNO administration. Moreover, the thermographic data indicated the presence of a decreased signal in the iBAT and an increase in tail temperature in the VGLUT2-Cre mice (Fig. 5e), which was in concordance with the response observed in mice exposed to the 2$g$ environment (Fig. 2a). Therefore, VGLUT2-expressing neurons located in the VNC possibly contributed to 2$g$-induced hypothermia, even though c-fos expression in the peripheral vestibular organ was observed in VGLUT2- and VGAT-expressing neurons in the VNC, which was induced by exposure to the 2$g$ environment (Supplementary Fig. 9).

**VGLUT2-expressing neurons in the VNC have a crucial role in 2$g$-induced hypothermia.** We used the viral vector, AAV2–DIO–taCasp3–TEVp, to delete VGLUT2- or VGAT-expressing neurons in VNC to examine whether VGLUT2-expressing neurons in the VNC contributed to 2$g$-induced hypothermia (Supplementary Fig. 10). The decrease in BT was significantly attenuated by deleting VGLUT2-expressing neurons in the VNC (Fig. 6a–c and Supplementary Fig. 10). Moreover, the activity in the 2$g$ environment was significantly higher in VGLUT2-expressing neurons-deleted mice than that in the

control group (Fig. 6d), which was similar to the VL mice (Fig. 1b, c). Conversely, the deletion of VGAT-expressing neurons in the VNC affected neither BT response, nor activity, during 2$g$ exposure (Fig. 6c, d). These data suggest that VGLUT2-expressing neurons in the VNC are indispensable in 2$g$-induced hypothermia. Deletion of either VGLUT2- or VGAT-expressing neurons located in the VNC did not affect the rotarod skill (Fig. 6e). A compensatory mechanism of the non-vestibular system might contribute to the maintenance of the rotarod skill, since deletion of the targeted neurons by the viral vector is a longer process[23].

Vestibular training using exercise[24] or GVS[25] is beneficial for rehabilitation of vestibular loss or adaptation to outer space. Therefore, we hypothesized that the use of chemogenetic stimulation of VGLUT2-expressing neurons present in the VNC may mimic vestibular training and that the hypothermic response may consequently be attenuated. BT in the VGAT-Cre mice returned to baseline levels within 240 min of CNO administration during the chemogenetic activation using AAV–CAG–FLEX–hm3D(Gq)–mCherry, while BT recovery in VGLUT2-Cre mice occurred in 9 h (Fig. 7a). Although hypothermia induced by 2$g$ exposure was significantly attenuated by

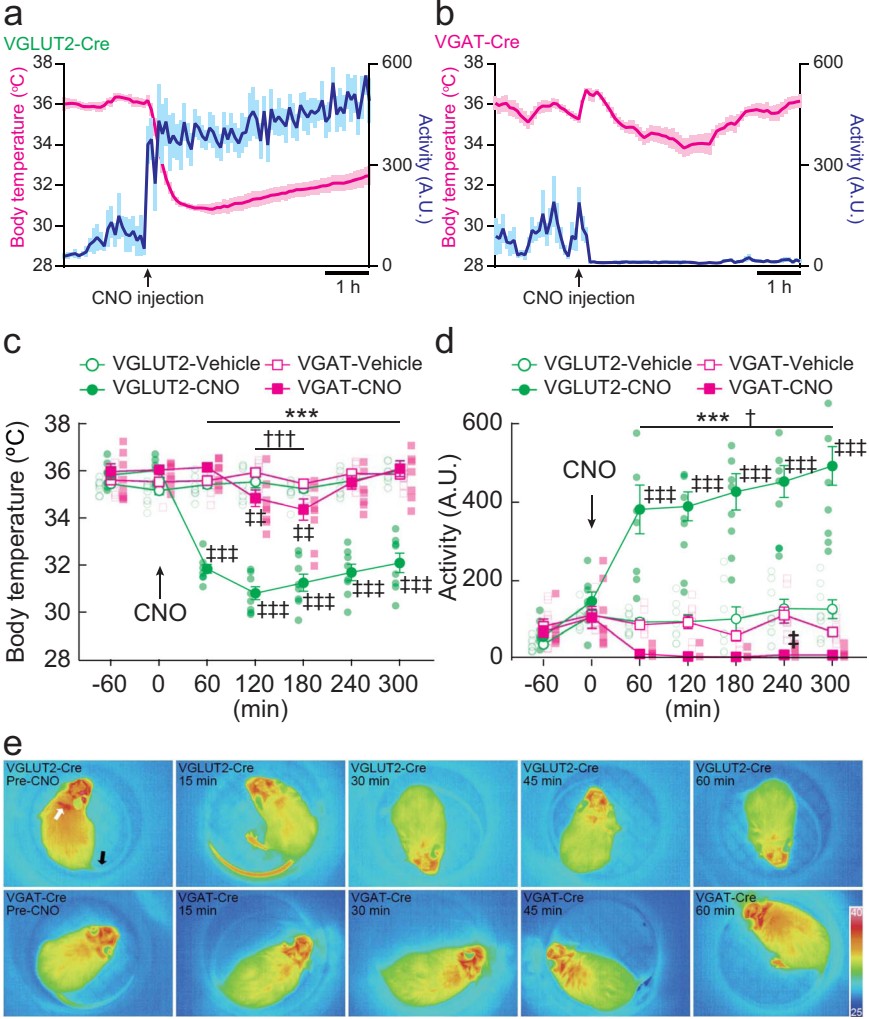

**Fig. 5 The role of VGLUT2- and VGAT-expressing neurons located in the vestibular nuclear complex (VNC) on thermoregulation. a**, **b** Averaged data of the body temperature (BT, magenta) and activity (blue) prior to and during clozapine N-oxide (CNO) administration in VGLUT2-Cre (**a**, $n = 8$) and VGAT-Cre mice (**b**, $n = 8$). The receptors for chemogenetics (i.e., hm3D(Gq)) were expressed in either VGLUT2- or VGAT-expressing neurons located in the VNC. The chemogenetic stimulation was applied in the bilateral VNC. **c**, **d** Summarized data of the changes in BT (**c**) and activity (**d**) induced by either CNO or its vehicle administration in VGLUT2-Cre and VGAT-Cre mice ($n = 8$ in each group). Two-way ANOVA with either the Tukey's or Dunnett's post-hoc test. * or † vs. averaged value of pre-loading in either VGLUT2-CNO or VGAT-CNO mice, respectively; ‡ vs. VGLUT2-vehicle or VGAT-vehicle. **e** Representative images of the changes in the interscapular brown adipose tissue (iBAT, white arrow) and tail (black arrow) temperature induced by the administration of clozapine N-oxide (CNO) in a VGLUT2-Cre and a VGAT-Cre mouse. A photograph was taken every 15 min using a thermographic camera. For all statistical analyses, single, double, or triple significant symbols indicate $P < 0.05$, $P < 0.01$, or $P < 0.001$, respectively. Detailed information on the statistical analyses is reported in Supplementary Table 2.

chemogenetic activation of VGLUT2-expressing neurons in the VNC, which was administered 2 days before the 2$g$ load (Fig. 7b, c), the same effect was not observed during the chemogenetic activation of VGAT-expressing neurons (Fig. 7c). Higher activity values were observed in mice with advanced chemogenetic activation of VGLUT2-expressing neurons in the VNC (Fig. 7d). Since BT recovery was achieved 9 h after CNO was injected to activate VGLUT2-expressing neurons (Fig. 7a), we hypothesized that the 2$g$ load for 9 h might attenuate the hypothermic response. Exposure to 2$g$ for 9 h, which was performed 2 days prior to the start of the experiment, significantly attenuated 2$g$-induced hypothermia (Fig. 7e, f). In contrast, neither exposure to the 2$g$ environment for 60 min, nor repeated exposure to it, attenuated 2$g$-induced hypothermia (Supplementary Fig. 11), suggesting that the duration, and nor the number of loading times, is important for protection from 2$g$-induced hypothermia.

## Discussion

This study demonstrated that otolith organs, and not semicircular canals, were involved in 2$g$ load-induced hypothermia. Hypothermia was caused by decreased heat production in the iBAT (peripheral mechanism), which was induced by a decrease in iBAT sympathetic nervous activity and/or plasma catecholamines. VGLUT2-expressing neurons (central mechanism), instead of VGAT-expressing neurons in the VNC, were responsible for 2$g$ load-induced hypothermia. Deletion or advanced training of these neurons could prevent the hypothermic response.

In the present study, decreased food intake and body mass were observed in addition to 2$g$ load-induced hypothermia. Both hypophagia and hypothermia are observed in motion sickness that is mediated by the vestibular system[26]. It is difficult to determine whether mice experience motion sickness because they show no emetic response. However, allotriophagy, an index of

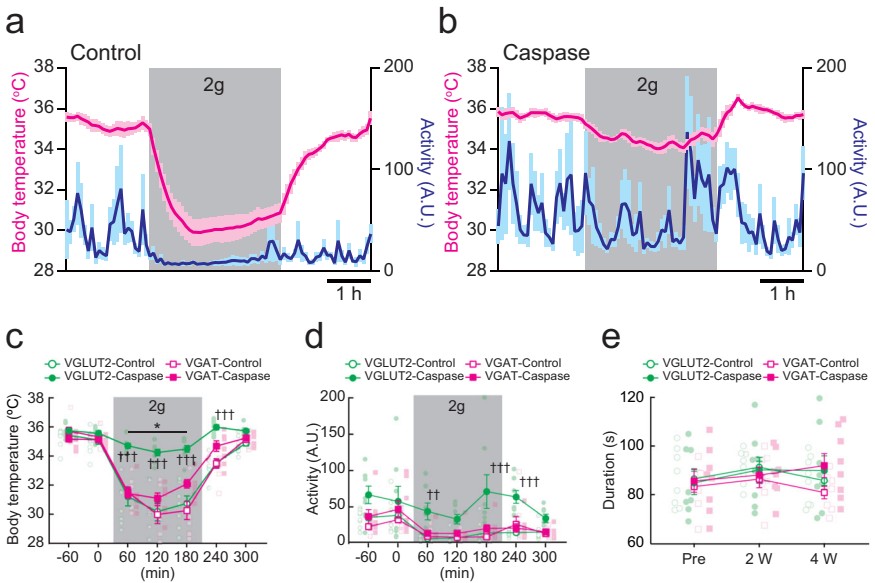

**Fig. 6 Role of VGLUT2-expressing neurons located in the vestibular nuclear complex (VNC) in 2g-induced hypothermia. a, b** Effect of deletion of VGLUT2-expressing neurons located in the VNC on 2g-induced hypothermia. Viral vectors, either AAV–DIO–EF1α–mCherry (**a**) or AAV2–DIO–taCasp3–TEVp (**b**), were injected in the bilateral VNC in VGLUT2-Cre mice ($n = 8$ in each group). **c, d** Summarized data of the changes in BT (**c**) and activity (**d**) induced by 2g load in VGLUT2-Cre and VGAT-Cre mice with the aforementioned viral vectors ($n = 8$ in each group). Two-way ANOVA with Tukey's post-hoc test. * vs. averaged value of pre-loading in all groups; † VGLUT2-caspase vs. VGLUT2-control. **e** Summarized data of the rotarod experiment following the injection of either the AAV2–DIO–taCasp3–TEVp or its control vector into the VNC in VGLUT2-Cre (i = 8) and VGAT-Cre ($n = 8$) mice. Each mouse underwent three sessions of the rotarod experiment, and the averaged value was used. For all statistical analyses, single, double, or triple significant symbols indicate $P < 0.05$, $P < 0.01$, or $i < 0.001$, respectively. Detailed information on the statistical analyses is reported in Supplementary Table 2.

motion sickness, was observed in rats under hypergravity[27,28]. The drug for motion sickness seemed to be effective for treating the motion sickness. Hypothermia induced by rotation, which stimulates the semicircular canals instead of the otolith organs, was suppressed by 5-HT3 receptor blockade[29]. Moreover, the involvement of the vestibular efferents, which terminate on the vestibular hair cells and release acetylcholine, is also suggested. Shaking-induced hypothermia was suppressed in the mice lacking the α9 acetylcholine receptor subunit that is predominantly expressed in the vestibular hair cells[30]. Furthermore, in humans, rotation with enhanced head movements decreases BT compared with rotation only[31]. The present study suggested that motion sickness mediated by the otolith organs and not the semicircular canals is more severe, which can explain the prolonged and severe symptoms observed in space motion sickness[32].

The 2g load-induced hypothermia observed in the present study might be attributed to a decrease in plasma catecholamines. The symptoms of motion sickness, including cold-sweating and pallor, are known to be induced by sympathoexcitation[33], which seems to contradict the present data. This might be attributed to the pattern of vestibular inputs, i.e., phasic or tonic inputs, since there are some reports that electrical stimulation of vestibular nerve elicits sympathetic nervous response including excitation, inhibation, or a combination of both[4]. Hammam et al., demonstrated that phasic vestibular inputs, such as sinusoidal GVS, changed skin sympathetic nerve activity: a large peak was associated with the positive peak of the sinusoid, and a smaller peak was associated with the negative phase[33]. On the other hand, tonic vestibular inputs, such as hypergravity (3g), increased renal sympathetic nerve activity in rats at the onset of loading; however, the value returned to the baseline level 3 min later[34]. Because short-term (4.5 s) microgravity exposure also induced sympathoexcitation[35], it is thought that sympathoexcitation might occur through the peripheral vestibular organ in response to

changes in vestibular inputs (phasic phase in gravitational change). In other words, sympathoinhibition and not sympathoexcitation might occur in case of long-term tonic vestibular inputs, which is supported by the evidence of previous studies[36,37].

The present study showed that advanced 2g load suppressed subsequent hypergravity-induced hypothermia. This training effect involves VGLUT2-expressing neurons in the VNC, because the chemogenetic activation of VGLUT2-expressing neurons was suppressed following 2g-induced hypothermia (Fig. 7c). Vestibular training seems to be effective against motion sickness[38], which requires selective activation of VGLUT2- instead of VGAT-expressing neurons in the VNC. GVS is an option for the activation of the neurons in the VNC[39]. Although the appropriate GVS pattern for activating VGLUT2-expresing neurons in the VNC is unknown, it can be used as a vestibular training tool for elderly patients with deteriorated motor function and even astronauts in space, if the GVS pattern can be optimized.

The present study showed that 2g load-induced hypothermia was suppressed by VL, suggesting that hypothermia is induced through the peripheral vestibular organ. Although the semicircular organs are also involved in 2g load-induced hypothermia as, reported by a previous study[40], the otolith organs might be the main modulators of thermoregulatory response (Supplementary Fig. 1). Vestibular and non-vestibular (visual, proprioceptive, and intestinal) inputs are required to understand the body's dynamics and kinematics[3]. In addition to vestibular inputs, non-vestibular inputs may also be involved in thermoregulation. Some intestinal signals are transmitted through the vagal afferents. Sympathetic nervous activity in the iBAT decreased depending on the increase in the frequency of electrical stimulation[41], suggesting that decrease in heat production through iBAT might occur in VL mice under the 2g load, because a small decrease was observed in

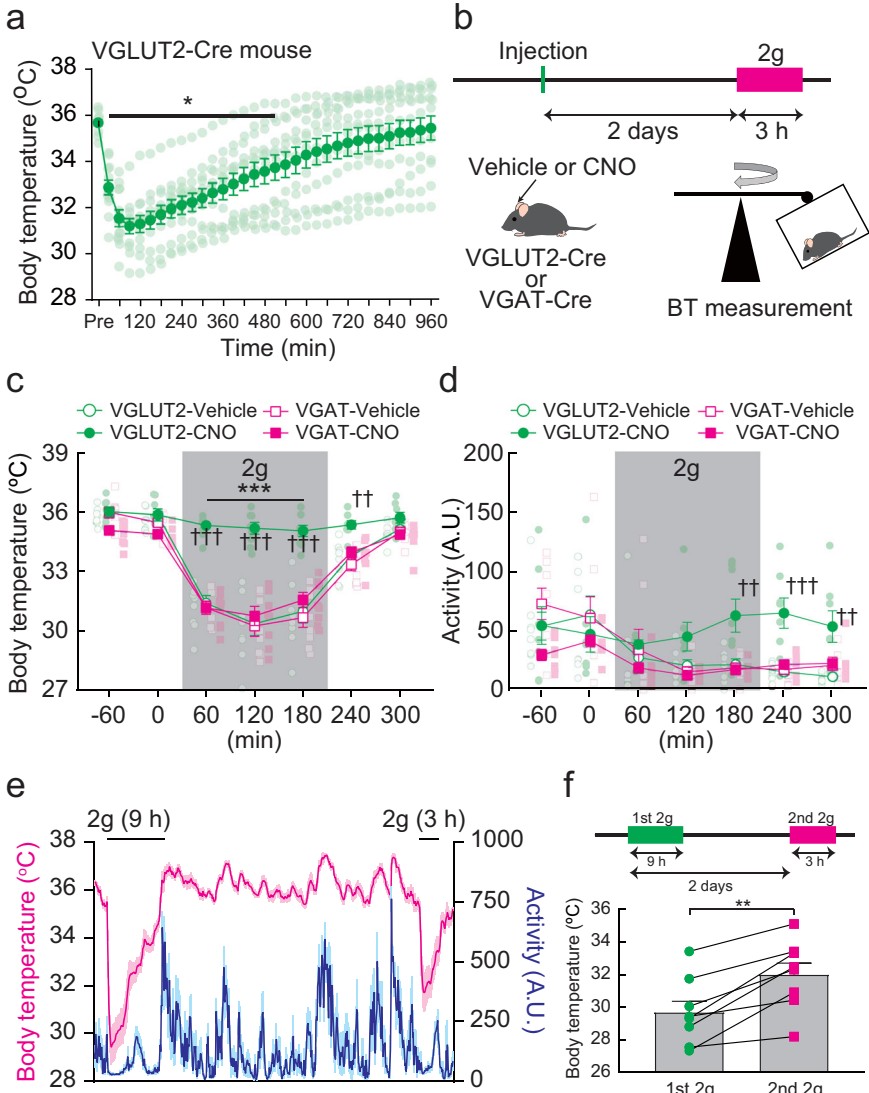

**Fig. 7 Exposure to the 2*g* environment attenuated the subsequent hypothermic response. a** Time-dependent changes in the body temperature (BT) following the injection of clozapine N-oxide (CNO) in VGLUT2-Cre mice ($n = 10$). The viral vector, AAV–CAG–FLEX–hm3D(Gq)–mCherry (AAV–hm3D (Gq)–mCherry), was injected in the VNC bilaterally. Each value was obtained by averaging the values every 30 min. One-way ANOVA with the Tukey's multiple comparisons test; *$P < 0.05$ vs. pre. **b** A schema of the assessment of the effect of CNO administration on the following 2*g*-induced hypothermia in VGLUT2-Cre and VGAT-Cre mice. **c, d** Summarized data of the changes in BT (**c**) and activity (**d**) induced by 2*g* load in VGLUT2-Cre and VGAT-Cre mice with either advanced CNO injection or delivery of its vehicle ($n = 8$ in each group). Two-way ANOVA with Tukey's post-hoc test. * vs. averaged value of pre-loading in VGLUT2-vehicle, VGAT-vehicle, and VGAT-CNO; † VGLUT2-vehicle vs. VGLUT2-CNO. **e** Exposure to 2*g* was applied for 9 h in C57BL/6 J mice ($n = 8$). The hypothermic response for 3 h was evaluated two days after the 2*g* experience. The averaged time-dependent changes in the BT (magenta line) and activity (blue line) are shown. **f** The effect of the 2*g* load on the following 2*g*-induced hypothermia in C57BL/6 mice. A paired t-test was applied for the statistical analysis. For all the statistical analyses, single, double, or triple significant symbols indicate $P < 0.05$, $P < 0.01$, or $P < 0.001$, respectively. Detailed information on the statistical analyses is reported in Supplementary Table 2.

BT (Fig. 1c). Hypothermia resulting from the vagal afferent requires signal transmission from the nucleus of the solitary tract (NTS) to the rostral raphe pallidus area, which contains iBAT sympathetic premotor neurons[17,18,41]. There is no projection from the VNC to the rostral raphe pallidus, magnus, or obscurus[42]. However, a neural projection was reported from the VNC to the NTS[43,44]. The present study showed that decrease in plasma catecholamine levels, which was probably caused by sympathoinhibition or decreased release of adrenaline from the adrenal gland, induces lower heat production from iBAT under 2*g* load. This response may be due to vestibular input-induced inhibition of neural activity in the rostral raphe pallidus area through the NTS.

The vestibular system contributes substantially to the stabilization of body posture during locomotion. Deletion of neurons in the lateral vestibular nucleus showed perturbation in the ipsilateral side[45]. Activation of neurons in the VNC induces contraction of the ipsilateral extensor muscle. Thus, unilateral photostimulation of VGLUT2-expressing neurons in the VNC should result in tilting of the body to the contralateral side; however, we observed that the body tilted to the ipsilateral side. Similarly, photoactivation of VGAT-expressing neurons in the VNC, which might be involved with the commissural inhibitory system that inhibits the ipsilateral VGLUT2-expressing neurons, induced body tilt to the contralateral side. In the spinal cord, communication between vestibular and proprioceptive signaling

is important for maintaining balance[46]. This communication is believed to ensure smooth motor behavior, i.e., proprioceptive reflex through the muscle spindle influences vestibular signaling to the muscle. Extension is accompanied with contraction of the extensor muscle and relaxation of the flexor muscle. It is possible that a transient stretch of the ipsilateral flexor muscle induced by photoactivation of VGLUT2-expressing neurons in the VNC induces the stretch reflex, and the body tilted to the ipsilateral side. These possibilities, including the mechanisms in the brainstem and proprioceptor, should be studied in the future.

The VNC is known to contain glutamatergic and GABAergic/glycinergic neurons[21]. The present study showed that VGLUT2-expressing neurons in the VNC participate in the $2g$ load-induced hypothermia. Chemogenetic activation of both unilateral and bilateral VGLUT2-expressing neurons in the VNC decreased BT. On the other hand, although chemogenetic activation of the unilateral VGAT-expressing neurons increased BT, bilateral activation slightly decreased BT (Fig. 5 and Supplementary Fig. 8). This might be because hypothermia is associated with a lower activity rather than because it is the direct effect of chemogenetic activation of the neurons, based on the following reasons: (1) Changes in BT were not observed at 60 min after CNO administration. Unilateral chemogenetic activation showed increase in BT at this time point. (2) A significant decrease in activity was observed during the bilateral activation for 5 h; however, unilateral activation did not change the activity. In the $2g$ environment, not only VGLUT2-expressing neurons in the VNC but also VGAT-expressing neurons were activated in accordance with the observed c-fos expression (Supplementary Fig. 9). Furthermore, lower activity of Sham mice compared with VL mice in the $2g$ environment seems to be due to activation of VGAT-expressing neurons (Fig. 1d), even though there was the same gravitational load in both groups. Accordingly, it is possible that hypothermia induced by $2g$ load is a result of the net effect, including the activation of VGLUT2-expressing neuron-mediated hypothermia and the activation of VGAT-expressing neuron-mediated lower activity, which decreases heat production. Taken together, VGAT-expressing neurons in the VNC participate in the BT response, although it is still unclear why only the unilateral activation or inhibition induced changes in BT.

The autonomic nervous system, notably its sympathetic division, is activated by physical and psychological stressors[47,48]. A subset of these autonomic responses is mediated or facilitated by the C1 neurons, a group of glutamatergic/catecholaminergic/peptidergic neurons present in the medullary reticular formation[49]. The C1 cells are anatomically heterogeneous[49], and subsets of these neurons operate as a switchboard for eliciting behaviorally appropriate patterns of sympathetic responses[49,50]. Previously, we demonstrated that the vestibular system participates in the sympathoexcitation followed by the pressor response[35,51]. The present study showed that a $2g$ load for 90 min induced c-fos expression in C1 neurons, which was significantly suppressed by VL (Supplementary Fig. 4), suggesting that C1 neurons are activated by inputs relayed from the peripheral vestibular organs. Interestingly, it seems that the outflow from C1 neurons differs for acute and chronic stress. Acute optogenetic stimulation of C1 neurons increased sympathetic nervous activity in rats[52]. In contrast, chronic optogenetic stimulation of C1 neurons showed a decrease in heart rate in mice[22], which was similar to the response observed in the present study (Supplementary Fig. 4). Accordingly, it is possible that decrease in both plasma catecholamines and heart rate induced by the $2g$ load is caused by chronic activation of C1 neurons. In the neural pathway, direct projections from the VNC to the rostral ventrolateral medulla, which contains C1 neurons, exist in rats[21]. On the other hand, some neurons in the VNC project to the NTS[49], which

could activate the neurons in the RVLM[44]. Furthermore, since hypergravity-induced increase in c-fos expression in the paraventricular hypothalamic nucleus was suppressed by VL[14], C1 neurons may be also influenced by neurons in the diencephalon[49]. Thus, it is possible that C1 neuronal activation by vestibular stimulation is the result of a direct or an indirect pathway.

Thyroid hormones have crucial roles in maintaining basal metabolic rate, thermogenesis, and lipid and carbohydrate metabolism, indicating that the hypothalamus–pituitary–thyroid axis is an important factor in energy homeostasis[53]. Neurons of endocrine thyrotrophin-releasing hormone in paraventricular nucleus receive information from C1 and NTS neurons[54]. Since C1 and NTS[14] neurons are activated by hypergravity, the hypothalamus–pituitary–thyroid axis may be involved in $2g$ load-induced hypothermia. Thyroid hormones seem to be important for maintaining BT in hypothermic mice[55]. Although there are no reports on the changes in plasma thyroid hormones under hypergravity, a collapse of thermoregulation in the $2g$ environment might not be the sole factor responsible for sympathoinhibition; hypothyroidism may also be an important factor, because administration of beta-adrenergic receptors agonists did not completely reduce hypothermia (Fig. 2e). This possibility should be examined by future studies.

Heat produced by the heart cannot be neglected particularly in thermoregulation in small mammals such as mice. The heart rate was decreased under $2g$ load in the present study (Supplementary Fig. 4), suggesting the possibility of cardiac inhibition. Monson et al. reported a decrease in BT from 38 to 30 °C under the hypergravity load ($2.1g$) in rats[8], which was similar to the response of the mice in the present study. Previously, we reported that an increase in arterial pressure instead of hypotension occurred during the hypergravity load in conscious rats[6,34]. The decrease in heart rate was caused by the baroreflex[34]. On the other hand, the pulse pressure increased[6,34], suggesting an increase in the stroke volume. Thus, even if the bradycardia was observed, the size of the cardiac work might be maintained, i.e., the thermogenesis from the heart does not decrease during $2g$ load. Moreover, cardiac inhibition seems to have a small effect on hypothermia because propranolol decreased BT by 0.5 °C in rats, although a decrease in heart rate was observed[56]. Therefore, sympathoinhibition rather than cardiac inhibition may contribute to hypothermia during $2g$ load.

In summary, we found that VGLUT2-expressing neurons in the VNC, which receive inputs from the otolith organs in the inner ear have a crucial role in $2g$ load-induced hypothermia. Stress responsive neurons, including C1 neurons, may participate in the hypothermia response. However, very little is understood about the physiological significance of thermoregulation by the vestibular system. Further studies are needed to examine these aspects.

## Methods

**Animals**. The animals used in the present study were maintained in accordance with the "Guiding Principles for Care and Use of Animals in the Field of Physiological Science", set by the Physiological Society of Japan. The experiments were approved by the Animal Research Committees of Gifu University. Male ($n = 90$) and female ($n = 86$) C57BL/6J mice, weighing 20–30 g, were purchased from Chubu Kagaku Shizai. In contrast, VGLUT2–Cre (STOCK Slc17a6$^{tm2(cre)Lowl/J}$) and VGAT–Cre (STOCK Slc32a1$^{tm2(cre)Lowl/J}$) mice were obtained from Jackson Laboratories and were maintained on a C57BL/6J background. A total of 84 VGLUT2–Cre (male, 42; female, 42) and 84 VGAT–Cre (male, 42; female, 42) mice, aged 8–14 weeks, were used for our experiments.

**Anesthesia and postoperative management**. All surgeries were conducted under aseptic conditions, and mice were anesthetized with a mixture of medetomidine hydrochloride (0.3 mg/kg), midazolam (4 mg/kg), and butorphanol tartrate (5 mg/kg, i.p.). The depth of anesthesia was deemed sufficient when the corneal and hindpaw withdrawal reflexes were absent. Additional anesthetic was administered as

necessary (10% of the original dose, i.p.). BT was maintained at 37.0 ± 0.5 °C with a servo-controlled temperature pad. After surgery, mice received postoperative boluses of atipemazole (an $\alpha_2$-adrenergic antagonist, 2 mg/kg, s.c.), penicillin G potassium (3000 U/kg, s.c.), and ketoprofen (4 mg/kg, s.c.). Mice were housed in groups of 4/cage under a 12:12 h light-dark cycle. The room temperature was maintained at 24 ± 1 °C.

**Vestibular lesion.** The surgery inducing the bilateral vestibular lesion (VL) was performed according to a previously described method[15]. After careful removal of the tympanic membrane, malleus, incus, and stapes, the labyrinthine fluid was aspirated. The probe of the ultrasonicator was placed beside the oval window and the sonication was applied. Conversely, while the tympanic membrane was also removed during the sham VL surgery, the auditory ossicles were left intact. About one week following both surgeries, the success of the operation was confirmed by observing the swimming behavior of the mice. They were placed on a sieve basket, which was gently placed in a small tub filled with warm water. In cases of complete lesions, the mice were unable to determine the direction in which they had to swim and continued to turn around under the water. The sieve basket was raised from the water immediately following observation of this behavior. No mice drowned or died as a consequence of the swimming test.

**Exposure to the hypergravity environment.** To expose mice to the 2$g$ environment, centrifugation of the gondola-type rotating box with either a 1.5 (Shimadzu) or 0.25 (The Japan Aerospace Exploration Agency (JAXA)) m arm was employed. The long arm centrifugation was used for the one-week-long experiment, whereas the short arm centrifugation was utilized for the other experiments. The 2$g$ environment was created within 70 s, except for the data in Fig. 1f. All the mice were able to access food and water ad libitum. The room temperature was maintained at 24 ± 1 degrees Celsius. In the case of the one-week-exposure, a 12:12 light-dark cycle was established. The behavior of the mice was recorded day and night, continuously. With regard to the on-axis rotation, a mouse was placed in an adequately ventilated 50-mL conical plastic tube (Corning Inc.), with his head on the axis of rotation to stimulate the semicircular organ only. The rpm of the on-axis rotation was the same as that of the 2$g$ experiment. In Fig. 1f, body mass and food intake were measured at the cessation of 2$g$, i.e., the data were from exposure to 1$g$ or 2$g$ environment for 48 h.

**Measurement of BT and activity.** The implantable programmable device for the measurement of BT and activity was used in our experiment (nano tag®, KISSEI COMTEC CO., LTD). The device was implanted in the abdominal cavity of the mice. One week after the surgery, the measurement time and sampling rate (5 min intervals) were programmed using a non-contact IC card. All the measurements started at 11 am, except for the data in Fig. 1a–d which were obtained at 7 am. All the data saved in the device were obtained through the same non-contact IC card after the experiment. In the case of the measurement of the interscapular brown adipose tissue (iBAT) temperature, the sensor of the device was placed on it. The appropriate placement of the iBAT was confirmed using a thermographic camera (TVS-200, Nippon Avionics Co., Ltd.). As a hypothermic response, the minimum value of the BT either during 2$g$ or following the clozapine N-oxide (CNO) administration was used for the data analysis except for those in Fig. 1c, d. Averaged values for 24 h of 2$g$ exposure were employed in Fig. 1c, d. The baseline temperature before 2$g$ load or CNO injection is shown as averaged values.

**Measurement of heart rate.** A radio-telemetry probe (ETA-F10; Data Sciences International) was implanted in a mouse to measure the electrocardiogram (ECG). ECG signals were obtained through a PhysioTel Receiver (RLA 1020; Data Science International). The signals were recorded using an analog-to-digital converter (PowerLab, ADinstruments) at a rate of 100 Hz, then the heart rate was calculated from the R-R interval of the ECG signals.

**Measurement of noradrenaline and adrenaline levels.** Blood was collected from the ophthalmic artery using a glass tube (Fisherbrand Microhematocrit Capillary Tubes, Fisher Scientific) under isoflurane inhalation. Isoflurane was given via the swivel-connected tube during 2$g$ load. Plasma noradrenaline and adrenaline were extracted using manufactural kits (MonoSpin® PBA, GLSciences) and were measured using high-performance liquid chromatography (HTEC-510, Eicom).

**Drugs.** All the drugs, except hexamethonium, were injected intraperitoneally using an implantable and programmable microinfusion pump system (iPRECIO Micro Infusion Pump System, SMP-300, Primetech Corp.) and a syringe with a 26 G needle. Right before implantation of the microinfusion pump, the infusion start and end times and the infusion rate (10 μL/h) were programmed using the provided software (iPRECIO IMS-300 Management Software, Primetech Corp.). To inject adequate doses of the drugs for 15 min, each drug solution was prepared to the appropriate concentration. Subsequently, the catheter part of the microinfusion pump was placed in the abdominal cavity, while the main body of the microinfusion pump was implanted s.c. in the lumbar region. Finally, hexamethonium was injected intraperitoneally using a syringe with a 26 G needle.

**Viral vectors.** AAV–DIO–EF1α–Channelrhodopsin2(H134R)–mCherry serotype 2 (AAV2–ChR2–mCherry), AAV–EF1a–DIO–eArch3.0–eYFP serotype 2 (AAV2–eArch–eYFP), AAV–DIO–EF1α–mCherry serotype 2 (AAV2–mCherry), AAV–DIO–EF1α–eYFP serotype 2 (AAV2–eYFP), and AAV2–DIO–taCasp3–TEVp (taCasp3–TEVp) were purchased from the University of North Carolina vector core [the first two constructs were obtained courtesy K. Deisseroth (Stanford University), while the third construct was obtained courtesy N. Shah (University of San Francisco)]. The AAV–CAG–FLEX–hm3D (Gq)–mCherry (AAV–hm3D(Gq)–mCherry) was obtained from the Nagoya University [the construct was obtained courtesy Akihiro Yamanaka]. The AAV–DIO–hSyn–hm4D(Gi)–mCherry (AAV–hm4D(Gi)–mCherry) was purchased from Addgene (https://www.addgene.org/). In these vectors, the ChR2–mCherry (AAV2–ChR2–mCherry), eArch–eYFP (AAV2–eArch–eYFP), mCherry (AAV2–mCherry), eYFP (AAV2–eYFP), taCasp3–TEVp, hm3D (Gq)–mCherry, and hm4D(Gi)–mCherry sequences are flanked by the same double lox sites (LoxP and lox 2722).

**Injection of the viral vectors and optical fiber placement.** The AAV2–ChR2–mCherry, AAV2–eArch–eYFP, AAV2–mCherry, AAV2–eYFP, or AAV–hm4D(Gi)–mCherry were injected unilaterally into the left VNC, which was followed by the placement of an optical fiber. In contrast, the AAV–hm3D (Gq)–mCherry was injected unilaterally or bilaterally into the VNC, whereas the taCasp3–TEVp was injected bilaterally into the VNC (Supplementary Fig. 5b). The VNC was located by mapping the caudal end of the facial motor nucleus, dorsal to which the VNC lies. The mandibular branch of the facial nerve was revealed through a small skin incision (either left side or both sides, as required) for successive electrical stimulation. Thereafter, the mouse was placed prone on a stereotaxic apparatus (SR-6M-HT, Narishige) adapted for mouse stereotaxic injections. The viral vector was loaded into a 1.2-mm internal diameter glass pipette broken to a 25-μm tip (external diameter) and introduced into the brain through a 1.5-mm diameter hole drilled into the occipital plate caudal to the parieto-occipital suture on the left side (or on both sides). The facial nerve was then stimulated (0.1 ms, 1–300 μA, 1 Hz) to evoke antidromic field potentials within the facial motor nucleus. These field potentials, recorded via the vector-filled pipette, were used to map the caudal end of the facial motor nucleus (FN) and identify the location of the VNC, which resides dorsal to the facial motor nucleus. Three 140 nL injections were performed 2000 μm above the base of the medulla oblongata (determined as the lower limit of the facial field potential). The three injections were separated by 200 μm and were aligned rostrocaudally. Successively, an optical fiber (125 μm core after desheathing; 0.39 numerical aperture; Thorlabs) was inserted 300 μm above the injection site and secured to the skull through a cyanoacrylate adhesive. Prior to the implantation, the optical fibers were glued to a zirconia ferrule (outside diameter, 1.25 mm; bore, 130 μm; Precision Fiber Products). The same stereotaxic procedure was used for the injection of either taCasp3–TEVp, AAV–hm3D(Gq)–mCherry, or AAV–hm4D(Gi)–mCherry into the VNC, even though optical fibers were not inserted. Mice were housed in groups of 4/cage under a 12:12 h light-dark cycle after surgery. The room temperature was maintained at 24 ± 1 °C. We used all mice for the experiments 4 weeks after surgery.

**Balance test using photostimulation.** A blue laser (MBL-III-473, Changchun New Industries Optoelectronics Technology Co. Ltd.) and a green laser (MBL-III-532, Changchun New Industries Optoelectronics Technology Co. Ltd.) were used to stimulate the ChR2 and eArch of the neurons located in the VNC, respectively. The light output of each optical fiber was measured with a light meter (PM20A, Thorlabs) and the laser setting required to deliver 10 mW was recorded. Mice were briefly anesthetized with isoflurane while the connection between the implanted fiber optic and the laser delivery system was established. After recovery from the isoflurane inhalation, a mouse was placed on the rod located 50 cm above the ground. The parameters related to photostimulation are as follows: duration, 10 ms; frequency, 10, 20, 30, 40 Hz. Photostimulation was also applied with a 1-s duration (hold). Body tilt was recorded during photostimulation using a video camera (EX-100F, CASIO), and its degree was calculated using software (https://www.kinovea.org/).

**Rotarod experiment.** The vestibular system-related coordinated movements were estimated by placing mice on the rotary rod (47600, Bioresearch Center) with their heads facing opposite to the direction of rotation. The rotation speeds increased successively from 2 to 40 rpm in 2 min after which the mice were required to move forward to remain on the rod. The time spent on the rotating rod was measured for each mouse.

**Tracking movement during chemogenetic stimulation.** A mouse was placed in a column-shaped box (diameter, 240 mm; height, 100 mm), while a video camera (EX-100F, CASIO) was positioned above the box. Following adaptation to the box for 30 min, the mouse's movements were recorded. After 10 min, either the receptor agonist CNO (C-929, NIMH Chemical Synthesis and Drug Supply Program, 3 mg/kg) or its vehicle (saline) were injected. Preliminary experiment showed that neither movement nor BT were affected by CNO itself (3 mg/kg) in C57BL/6 J

mice. The data from the recording video were transferred to our personal computer for the analysis of the movement distance (TopScan, CleverSys).

**Immunohistochemistry**. Mice were euthanized with an overdose of pentobarbital sodium and perfused transcardially with 50 mL of heparinized saline, followed by 100 mL of freshly prepared 4% paraformaldehyde in 100 mL sodium phosphate buffer (pH 7.4). Subsequently, their brains were extracted and post-fixed at 4 °C for 24–48 h in the same fixative. Transverse sections (40 μm thick) were then cut via a cryotome and stored in a cryoprotectant solution (20 % glycerol plus 30% ethylene glycol in 50 mM phosphate buffer, pH 7.4) at −20 °C. To confirm the expression of either the mCherry or eYFP in the VNC, the following antibodies were used: the mCherry protein was detected with the anti-DsRed (rabbit polyclonal, 1:500; Clontech #632496; Clontech Laboratories) followed by the Alexa Fluor-594-tagged donkey anti-rabbit antibody (1:200; Jackson ImmunoResearch Laboratories); the eYFP protein was detected with the anti-GFP (chicken polyclonal, 1:500; GFP-1010; Aves Labs) followed by the Alexa Fluor-488-tagged rabbit anti-chicken antibody (1:200; Jackson ImmunoResearch Laboratories). To examine the c-fos expression, the anti-c-fos (1:1,000; Millipore #ABE457; EMD Millipore) was used, followed by the Alexa Fluor-488 or 594-tagged donkey anti-rabbit antibody (1:200; Jackson ImmunoResearch Laboratories). To detect the tyrosine hydroxylase (TH)-expressing neurons, the anti-TH antibody (1:1,000; Millipore #AB1542; EMD Millipore) was used, followed by an Alexa Fluor-488-tagged donkey anti-sheep antibody (1:200; Jackson ImmunoResearch Laboratories). Thereafter, the brain sections were analyzed using fluorescence microscopy (BZ-X800, KEYENCE). The output levels were adjusted to include all the information-containing pixels, while the balance and contrast were adjusted to reflect the true rendering as much as possible. No other image retouching was performed.

**In situ hybridization**. The RNAscope Multiplex Fluorescent Assay kit (Advanced Cell Diagnostics) was used. Sections were washed in a sterile sodium phosphate buffer, mounted on charged slides, and dried overnight. All sections were mounted and reacted on the same slide for an experimental "run"; therefore they experienced the same experimental conditions and solutions. Following two rinses in sterile water, all sections were incubated with the "protease 4" for 30 min at 42 °C and then rinsed twice in sterile water. Subsequently, they were incubated in the RNAscope catalog oligonucleotide probes for VGAT (vesicular GABA transporter, Slc32a1, NM_009508.2) and VGLUT2 (vesicular glutamate transporter 2, Slc17a6, NM_080853.3) mRNA transcripts for 2 h at 40 °C. Finally, the tissue was treated according to the manufacturer's protocol following incubation in probes.

**Statistics and reproducibility**. All the data sets were tested for normality using either the D'Agostino–Pearson omnibus normality or Kolmogorov–Smirnov tests. Equal variances were examined successively through the Brown-Forsythe test. If the criteria of normality and equal variance were satisfied, the statistical significance was evaluated using either one- or two-way ANOVA, followed by either the Tukey–Kramer, Dunnet or Bonferroni tests. All values were expressed as means ± SEM, while the statistical significance was set at $P < 0.05$. The reproducibility of the experiments is in the Reporting Summary.

**Reporting summary**. Further information on research design is available in the Nature Research Reporting Summary linked to this article.

## Data availability
The authors declare that all data supporting the findings of this study are available within the article and its supplementary information files. All source data underlying the graphs presented in the main or supplementary figures are made available as Supplementary Data 1.

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

## Acknowledgements

The editorial comments provided by Patrice G. Guyenet, PhD (University of Virginia, Pharmacology Department) are gratefully acknowledged. Special thanks to the Grant-in-Aid for Scientific Research on Innovative Areas 15H05940 (HM) and 18H04974 (CA), Grant-in-Aid for Scientific Research (C) 18K06850 (HM) and 19K07283 (CA), JST CREST JPMJCR1656 (AY), The Takeda Science Foundation (CA), The Uehara Memorial Foundation (CA), Kato Memorial Bioscience Foundation (CA), The Nakatomi Foundation (CA), Mochida Memorial Foundation for Medical and Pharmaceutical Research (CA), Daiichi Sankyo Foundation of Life Science (CA), Japan Aerospace Exploration Agency (HM) and Gifu University (CA).

## Author contributions

C.A., Y.Y., and H.M. designed the study protocols; C.A., Y.Y., Y.M., T.M., S.Y., A.Y. and H.M. conducted experiments and acquired and analyzed the data; and C.A. and H.M. wrote the manuscript.

## Competing interests

The authors declare no competing interests.
