## [Peer Review File · Communications Biology]

Reviewers' comments:

Reviewer #1 (Remarks to the Author):

In this very interesting paper by Abe, Yamaoka and colleagues, the authors report that glutamate-releasing cells of the brainstem vestibular nuclear complex mediate the hypothermic response to hypergravity exposure. The experimental work appears to have been executed at a high standard and I find the results compelling and exciting. This paper also appears to be among the first to deploy more contemporary neuroscience "tools" in the domain of gravitational biology. I have a few critiques to offer, all of which are intended to be constructive, and once appropriately addressed will make this good paper a great paper.

1. I think the Introduction could benefit from a restructure. I say this because the current Introduction fails to provide the rationale for the study, which instead is provided in the initial portion of the Results section. My specific critique is that it is entirely unclear to the naïve reader (as this Reviewer was) why the authors would think vestibular signaling was linked with temperature regulation. After spending some time reading the literature, in particular Reference #10 (Fuller et al., PNAS, 2002), I better understand why the authors would propose this hypothesis. My strong suggestion is for the authors to reframe their Introduction so that the reader is aware of this prior (and apparently quite seminal) work linking the vestibular system, in particular the otolith organs, with gravity-mediated changes in temperature regulation. Doing so would provide a strong rationale for the genetically-driven experimental work on the VNC in the current study. In addition, now that I am more familiar with this literature, my suggestion is also one of scholarship – it is important to provide appropriate attribution (i.e., cite) of prior studies that have informed current work.

2. Related to the foregoing, I also came across another paper by the same authors of Ref #10 that described a markedly different pattern of c-Fos activation in otolith deficient mice compared with wildtype mice following acute hypergravity (2G) exposure (Fuller et al., Neuroscience, 2004). This paper further described differential cellular activation responses in preoptic and hypothalamic regions associated with temperature and autonomic regulation. This paper should clearly be cited.

3. In the results the authors describe the effect of on- v off-axis rotations on body temperature, concluding that the otolith organs are the dominant sensory-transduction element driving the G-mediated temperature responses. I would suggest that the authors append the sentence that ends on line 99 on Page 4, ", as was previously shown by Fuller and colleagues (Ref #10)".

4. I was unable to find the dose of CNO used in the hm3Dq experiments in the paper. This is important information to provide given non-specific effects of CNO at higher doses.

5. Contrasting Figure 1A v 1B suggests a potential influence on the regulation of the body temperature rhythm as well. This deserves comment, in particular in light of other studies describing a potential link between circadian regulation and the otolith organs (another paper I discovered in my literature search! i.e., Fuller and Fuller, 2006).

6. I was unable to find the injection volumes for the AAVs used. I suspect that they were quite large given the low tropism of the AAV2 serotype used. Please provide this information.

7. I am confused why the authors used an immunohistochemical approach to determine 'deletion' of VGLUT2+ cell bodies in the VNC by the taCasp3-AAV. Theoretically this would require in situ analysis as VGLUT2 protein is typically not detectable in somas, but rather only in the terminals of VGLUT2-expressing neurons. Did the authors confirm that their VGLUT2 Ab is capable of revealing VGLUT2+

somas? Supp Figure 10 does not look like somal staining.

8. How long did the authors wait prior to testing (+G, Rotarod, etc) the mice with taCasp3-AAV injections? The taCasp3 toxin produces a rather slowly developing lesion. Please add information on timing (i.e., weeks post AAV injection) for all testing on all mice.

9. The fusion fluorophore appears to be different for the hm3Dq injected into the Vgat-cre and Vglut2-cre mice? Was a different construct used? If so, why?

Reviewer #2 (Remarks to the Author):

The work of Abe and col. has demonstrated the neurophysiological mechanism of hyper G related hypothermia through the vestibular glutamatergic system mainly the otolithic organs. Moreover, the chemogenetic approach provide very interesting data in term of vestibular lesion/compensation. Huge amount fo work. The results are excellent and the methodology properly performed.

Could you better link/justify the Glut activation related to the otolithic organs more than the semi-circular canals.

I agree that the otolithic system is probably the main modulator of thermoregulatory response however Ossenkopp an col. 1994 have reported hypothermia following rotation in rat. You could ad dit in the discussion.

Heat production is generated by BAT of course as shown here, but also by cellular metabolism and thyroid hormones. Did you measure it ? Does BAT inhibition is the only factor of hypothermia induced bu hyperG ? If not could you please mention/discuss it in the general discussion.

More generally, you should discuss that : does the vestibular system induces hypothermia through a specific neuronal route directly connected to the vegetative center and BAT/adrenergic system (specific response related to hyperG/high level motion ?) or a more general response to the anterior or posterior nucleus of the hypothalamus that in turn modulate the BAT and adrenergic response as well as hormonal (hypothalamo-hypophysis axis : thyroid mainly) and cellular metabolim level too?

I have some specific comments below.

Specific comments :

Abstract : the term stressor appears to me a bit confusing.

Gravity is one of the fundamental force with which all living species have been developped. I agree that it is a stressor in term of cardiovascular challenge, but it is a physical parameter that support the psychomotor development and numerous cognitive functions at adulthood (spatial cognition).

Gravity is not only a « stressor » but a crucial parameter for living species.

The other part is clear.

Introduction :

You mention : « The peripheral gravity sensor, located in the inner ear, consists of two components, namely the semicircular canals and the otolith organs, which detect angular and linear accelerations »

I agree with this part (dynamic sensor) but you shuold add that the otolithic organ also detect head position (static sensor) according to the gravitationnal force.

You mention : « Accordingly, it is possible that the stress via the peripheral vestibular organs induces hyperthermia as same as the other stress including air jet, restraint, social defeat, novel cage, cage

switch, and handling ». However, vestibular stimulation induces hypothermia has ever been reported in previous studies (Fuller and col 2002, 2004 in mice devoided of otoliths, Martin et al. 2015 in vestibulo-lesioned rat).

=>Please change this sentence

Please add a small paragraph on what we know between vestibulo-hypothalamic and vestibulo-vegetative centers neuronal routes. The Vest. System is a powerful modulator of the vegetative centers (please refer to Yates and col. firstly).

You mention : « In this study, we found that hypergravity-induced hypothermia might be a good model to clarify the mechanism in stress-induced hypothermia »

=>I am not agree with you. Hypergravity is probably a strong « unnatural » stressor, and a specific motion/gravity related stressor. The question is that : hypergravity stressor induce a general unspecific stress reponse or a specific one related to the vestibular sensor of motion/gravity. Please change this sentence.

You mention : « We elucidated the peripheral and central mechanisms of decrease in BT induced by vestibular stimulations including hypergravity or chemogenetic tools, which manipulate the neurons in the central nervous system »

=>You have elucidated the neurophysiological mechanism of hypergravity induced hypothermia through a chemogenetic approach.

You mention « creation of » : I suggest « exposure »

Results :

You mention « suggesting that the degree of motion sickness (estimated by decrease in BT and food intake) might be suppressed by slow increase in gravity from 1g to 2g. »

=>The biological sign of hypothermia as a marker of motion sickness is interesting and should be presented in the introduction.

You mention : « useful tool for training and that the hypothermic response may consequently be attenuated », do you suggest it for therapeutics use in the future ? It appears to me a bit speculative and hazardous. Please moderate your sentence.

Discussion :

You mention : « In other words, sympathoinhibition rather than sympathoexcitation might occur in cases of long-term tonic vestibular inputs, as observed in the present study. »

I agree with you, you might mention the long term vestibulo-sympathetic-bone regulation where vestibular inputs inhibit/reduce the sympathetic activity within bones of lower limb.

Vignaux G, Besnard S, Denise P, Elefteriou F. The Vestibular System: A Newly Identified Regulator of Bone Homeostasis Acting Through the Sympathetic Nervous System. *Curr Osteoporos Rep.* 2015 Aug;13(4):198-205. doi: 10.1007/s11914-015-0271-2. Review.

Vignaux G, Besnard S, Ndong J, Philoxène B, Denise P, Elefteriou F. Bone remodeling is regulated by inner ear vestibular signals. *J Bone Miner Res.* 2013 Oct;28(10):2136-44.

Moreover according to the medullary level, the vestibulo-sympathetic response changes stimulating or inhibiting (see Yates and col.)

Methods :

Injections seem to be properly performed according to Fig3 even if pictures focus on the Vestibular

Nucleus area.

Reviewer #3 (Remarks to the Author):

The authors of the MS "Role of glutamatergic neurons in the vestibular nuclear complex on the gravitational stress-induced hypothermia in mice" analysed gravitational stressors that activated the peripheral vestibular organs, thereby inducing a sympathetic nervous response. The stressor can change body temperature. Long-term exposure to hypergravity induces hypothermia; the mechanism remains unknown. Peripheral and the central mechanisms of 2g load-induced hypothermia were identified in mice. The efferent mechanism indicated the involvement of sympathoinhibition that resulted in decreased heat production through the brown adipose tissue. 2g load-induced hypothermia resulted from the sympathoinhibition via the activation of VGLUT2-expressing neurons in the vestibular nuclear complex.

The research topic is original and can be interesting for a wide cohort of scientists not only in space-related area but also for those involved in therapeutic hypothermia treatment.

Despite the potential importance and broad auditory of readers, the MS in the current form cannot be published and needs major revision through substantial rewriting.

Major concerns

It is very difficult to follow the MS logic because of grammar mistakes and poor English style. So, they should be corrected.

The MS is absolutely unfocused. No logic in the literature and experimental data description. Even in the abstract the readers should jump from adipose tissue and heat production to vesicular glutamate transporters.

In general, the MS looks as a mix of different information.

Lines 68-69 "Furthermore, other forms of stress such as food deprivation¹⁴, hypoxia¹⁵, and rotation¹⁶ led to torpor and decrease in oxygen consumption..." It means from this sentence that hypoxia ...led to.. a decrease in oxygen consumption??? It is clear because of reduced oxygen supplementation.

Line 82 Hypothermia induced by 2g load is due to decreased plasma catecholamine level " but no info about catecholamine in the abstract.

Lines 83-88 This part is rather introduction but not the results

Also, major concern is that all changes in 2g load-induced hypothermia can be a result of hypoxia. As it was shown that hypergravity loading was accompanied with hypoxia (Borisova and Krisanova 2008, *Advances in Space Research* 42).

206 "activity in the 2g environment was significantly higher in VGLUT2-expressing neurons-deleted mice than in the control group" and this fact can be explained by a decrease in amount of excitotoxic glutamate during 2g-induced hypoxia.

217-219 "Accordingly, we hypothesized that use of chemogenetic stimulation of VGLUT2-expressing neurons present in the VNC is a useful tool for training and that the hypothermic response may

consequently be attenuated." But it has been suggested that hypothermia is a useful compensatory mechanism.

230-231 "Exposure to the 2g for 9 h, which was applied 2 days prior to the start of the experiment, significantly attenuated the 2g- induced hypothermia" .This fact can also indicate hypoxia preconditioning training.

Reviewers' comments:

Reviewer #1 (Remarks to the Author):

In this very interesting paper by Abe, Yamaoka and colleagues, the authors report that glutamate-releasing cells of the brainstem vestibular nuclear complex mediate the hypothermic response to hypergravity exposure. The experimental work appears to have been executed at a high standard and I find the results compelling and exciting. This paper also appears to be among the first to deploy more contemporary neuroscience “tools” in the domain of gravitational biology. I have a few critiques to offer, all of which are intended to be constructive, and once appropriately addressed will make this good paper a great paper.

1. I think the Introduction could benefit from a restructure. I say this because the current Introduction fails to provide the rationale for the study, which instead is provided in the initial portion of the Results section. My specific critique is that it is entirely unclear to the naïve reader (as this Reviewer was) why the authors would think vestibular signaling was linked with temperature regulation. After spending some time reading the literature, in particular Reference #10 (Fuller et al., PNAS, 2002), I better understand why the authors would propose this hypothesis. My strong suggestion is for the authors to reframe their Introduction so that the reader is aware of this prior (and apparently quite seminal) work linking the vestibular system, in particular the otolith organs, with gravity-mediated changes in temperature regulation. Doing so would provide a strong rationale for the genetically-driven experimental work on the VNC in the current study. In addition, now that I am more familiar with this literature, my suggestion is also one of scholarship – it is important to provide appropriate attribution (i.e., cite) of prior studies that have informed current work.

Thank you very much for your constructive comments. We totally revised the text in the introduction as a reviewer suggests. We also added appropriate citations in the text.

2. Related to the foregoing, I also came across another paper by the same authors of Ref #10 that described a markedly different pattern of c-Fos activation in otolith deficient mice compared with wildtype mice following acute hypergravity (2G) exposure (Fuller et al., Neuroscience, 2004). This paper further described differential cellular activation responses in preoptic and hypothalamic regions associated with temperature and autonomic regulation. This paper should clearly be cited.

Thank you very much for your comments. We included the information about the presympathetic neurons in the hypothalamus with a reference which a reviewer suggested

(line 78-94).

3. In the results the authors describe the effect of on- v off-axis rotations on body temperature, concluding that the otolith organs are the dominant sensory-transduction element driving the G-mediated temperature responses. I would suggest that the authors append the sentence that ends on line 99 on Page 4, “, as was previously shown by Fuller and colleagues (Ref #10)”.

Thank you very much for your comments. We appended the sentence as a reviewer suggested (line 120).

4. I was unable to find the dose of CNO used in the hm3Dq experiments in the paper. This is important information to provide given non-specific effects of CNO at higher doses.

Thank you very much for your comments. The information is in the method section (line 929-930).

5. Contrasting Figure 1A v 1B suggests a potential influence on the regulation of the body temperature rhythm as well. This deserves comment, in particular in light of other studies describing a potential link between circadian regulation and the otolith organs (another paper I discovered in my literature search! i.e., Fuller and Fuller, 2006).

Thank you very much for your comments. We appended the sentence as a reviewer suggested (line 114-115).

6. I was unable to find the injection volumes for the AAVs used. I suspect that they were quite large given the low tropism of the AAV2 serotype used. Please provide this information.

Thank you very much for your comments. The information is in the method part (line 889). We used same method of the previous study (Abe et al., Nat Neurosci, 2017).

7. I am confused why the authors used an immunohistochemical approach to determine ‘deletion’ of VGLUT2+ cell bodies in the VNC by the taCasp3-AAV. Theoretically this would require in situ analysis as VGLUT2 protein is typically not detectable in somas, but rather only in the terminals of VGLUT2-expressing neurons. Did the authors confirm that their VGLUT2 Ab is capable of revealing VGLUT2+ somas? Supp Figure 10 does not look like somal staining.

Thank you very much for your constructive comments. We did the additional experiments to examine whether VGLUT2- or VGAT-expressing neurons were deleted by the injection of the taCasp3-TEVp. Numbers of VGLUT2-expressing neurons in VNC was significantly suppressed by injection of taCasp3-TEVp in VGLUT2-Cre mice, while the numbers of VGAT-expressing neurons were maintained. On the other hand, VGAT-expressing neurons were specifically deleted in VGAT-Cre mice. These data were shown in the Supplementary Figure 10.

8. How long did the authors wait prior to testing (+G, Rotarod, etc) the mice with taCasp3-AAV injections? The taCasp3 toxin produces a rather slowly developing lesion. Please add information on timing (i.e., weeks post AAV injection) for all testing on all mice.

Thank you very much for your comments. We set 4 weeks after injection. We added the information in the text (line 899-901).

9. The fusion fluorophore appears to be different for the hm3Dq injected into the Vgat-cre and Vglut2-cre mice? Was a different construct used? If so, why?

Thank you very much for your comments. We used same construct of AAV virus.

Reviewer #2 (Remarks to the Author):

The work of Abe and col. has demonstrated the neurophysiological mechanism of hyper G related hypothermia through the vestibular glutamatergic system mainly the otolithic organs. Moreover, the chemogenetic approach provide very interesting data in term of vestibular lesion/compensation. Huge amount for work. The results are excellent and the methodology properly performed.

Could you better link/justify the Glut activation related to the otolithic organs more than the semi-circular canals. I agree that the otolithic system is probably the main modulator of thermoregulatory response however Ossenkopp an col. 1994 have reported hypothermia following rotation in rat. You could add it in the discussion.

Thank you very much for your comments. We added the text in the discussion with citation which a reviewer suggested (line 315-317).

Heat production is generated by BAT of course as shown here, but also by cellular metabolism and thyroid hormones. Did you measure it? Does BAT inhibition is the only factor of hypothermia induced by hyperG? If not could you please mention/discuss it in the general discussion.

Thank you very much for your constructive comments. We have never measured plasma thyroid hormones during 2g load. Since neurons of endocrine thyrotrophin-releasing hormone in paraventricular nucleus receives information from C1 and NTS neurons, it is possible that hypothalamus–pituitary–thyroid axis might be involved in 2g-induced hypothermia. We added this possibility in the text (line 403-414).

More generally, you should discuss that : does the vestibular system induces hypothermia through a specific neuronal route directly connected to the vegetative center and BAT/adrenergic system (specific response related to hyperG/high level motion ?) or a more general response to the anterior or posterior nucleus of the hypothalamus that in turn modulate the BAT and adrenergic response as well as hormonal (hypothalamo-hypophysis axis : thyroid mainly) and cellular metabolism level too?

Thank you very much for your constructive comments. As we mention above, we added the possibility about participation of hypothalamus–pituitary–thyroid axis on thermoregulation during 2g load in the text (line 403-414).

Specific comments :

Abstract : the term stressor appears to me a bit confusing.

Gravity is one of the fundamental force with which all living species have been developed. I agree that it is a stressor in term of cardiovascular challenge, but it is a physical parameter that support the psychomotor development and numerous cognitive functions at adulthood (spatial cognition).

Gravity is not only a « stressor » but a crucial parameter for living species.

The other part is clear.

Thank you very much for your comments. We changed the text in the abstract.

Introduction :

You mention : « The peripheral gravity sensor, located in the inner ear, consists of two components, namely the semicircular canals and the otolith organs, which detect angular and linear accelerations » I agree with this part (dynamic sensor) but you should add that the otolithic organ also detect head position (static sensor) according to the gravitationnal force.

Thank you very much for your comments. We revised the text as a reviewer suggest (line 54-55).

You mention : « Accordingly, it is possible that the stress via the peripheral vestibular organs induces hyperthermia as same as the other stress including air jet, restraint, social defeat, novel cage, cage switch, and handling ». However, vestibular stimulation induces hypothermia has ever been reported in previous studies (Fuller and col 2002, 2004 in mice devoided of otoliths, Martin et al. 2015 in vestibulo-lesionned rat).

=>Please change this sentence

Thank you very much for your comments. The sentence was deleted because we reconstructed the introduction.

Please add a small paragraph on what we know between vestibulo-hypothalamic and vestibulo-vegetative centers neuronals routes. The Vest. System is a powerful modulator of the vegetative centers (please refer to Yates and col. firstly).

Thank you very much for your comments. We made a small paragraph what we know between vestibulo-hypothalamic and vestibulo-vegetative centers neural routes (line 78-94).

You mention : « In this study, we found that hypergravity-induced hypothermia might be a good model to clarify the mechanism in stress-induced hypothermia »

=>I am not agree with you. Hypergravity is probably a strong « unnatural » stressor, and

a specific motion/gravity related stressor. The question is that : hypergravity stressor induce a general unspecific stress reponse or a specific one related to the vestibular sensor of motion/gravity. Please change this sentence.

Thank you very much for your comments. The sentence was deleted because we reconstructed the introduction.

You mention : « We elucidated the peripheral and central mechanisms of decrease in BT induced by vestibular stimulations including hypergravity or chemogenetic tools, which manipulate the neurons in the central nervous system »

=>You have elucidated the neurophysiological mechanism of hypergraity induced hypothermia through a chemogenetic approach.

Thank you very much for your comments. We changed the text. (line 98)

You mention « creation of » : I suggest « exposure »

Thank you very much for your comments. We changed the text. (line 123 and 125)

Results :

You mention « suggesting that the degree of motion sickness (estimated by decrease in BT and food intake) might be suppressed by slow increase in gravity from 1g to 2g. »

=>The biological sign of hypothermia as a marker of motion sickness is interesting and should be presented in the introduction.

Thank you very much for your comments. It seems that the sentence including motion sickness is not appropriate in the reframed introduction. The paragraph about the biological sign of hypothermia as a marker of motion sickness is still in discussion part.

You mention : « useful tool for training and that the hypothermic response may consequently be attenuated », do you suggest it for therapeutics use in the future ? It appears to me a bit speculative and hazardous. Please moderate your sentence.

Thank you very much for your comments. We are sorry for your confusing. We revised the text (line 240).

Discussion :

You mention : « In other words, sympathoinhibition rather than sympathoexcitation might occur in cases of long-term tonic vestibular inputs, as observed in the present study. »

I agree with you, you might mention the long term vestibulo-sympathetic-bone regulation where vestibular inputs inhibit/reduce the sympathetic activity within bones of lower limb.

Vignaux G, Besnard S, Denise P, Elefteriou F. The Vestibular System: A Newly Identified Regulator of Bone Homeostasis Acting Through the Sympathetic Nervous System. *Curr Osteoporos Rep.* 2015 Aug;13(4):198-205. doi: 10.1007/s11914-015-0271-2. Review.

Vignaux G, Besnard S, Ndong J, Philoxène B, Denise P, Elefteriou F. Bone remodeling is regulated by inner ear vestibular signals. *J Bone Miner Res.* 2013 Oct;28(10):2136-44.

Moreover according to the medulary level, the vestibulo-sympathetic response changes stimulating or inhibiting (see Yates and col.)

Thank you very much for your comments. We revised the text with appropriate citations.
(line 299)

Methods :

Injections seem to be properly performed according to Fig3 even if pictures focus on the Vestibular Nucleus area.

Thank you very much for your comments.

Reviewer #3 (Remarks to the Author):

The authors of the MS “Role of glutamatergic neurons in the vestibular nuclear complex on the gravitational stress-induced hypothermia in mice” analyzed gravitational stressors that activated the peripheral vestibular organs, thereby inducing a sympathetic nervous response. The stressor can change body temperature. Long-term exposure to hypergravity induces hypothermia; the mechanism remains unknown. Peripheral and the central mechanisms of 2g load-induced hypothermia were identified in mice. The efferent mechanism indicated the involvement of sympathoinhibition that resulted in decreased heat production through the brown adipose tissue. 2g load-induced hypothermia resulted from the sympathoinhibition via the activation of VGLUT2-expressing neurons in the vestibular nuclear complex. The research topic is original and can be interesting for a wide cohort of scientists not only in space-related area but also for those involved in therapeutic hypothermia treatment. Despite the potential importance and broad auditory of readers, the MS in the current form cannot be published and needs major revision through substantial rewriting.

Major concerns

It is very difficult to follow the MS logic because of grammar mistakes and poor English style. So, they should be corrected. The MS is absolutely unfocused. No logic in the literature and experimental data description. Even in the abstract the readers should jump from adipose tissue and heat production to vesicular glutamate transporters.

Thank you very much for your comments. We revised the manuscript then we asked professional English editing service to check the manuscript again.

In general, the MS looks as a mix of different information.

Lines 68-69 “Furthermore, other forms of stress such as food deprivation¹⁴, hypoxia¹⁵, and rotation¹⁶ led to torpor and decrease in oxygen consumption...” It means from this sentence that hypoxia ...led to.. a decrease in oxygen consumption??? It is clear because of reduced oxygen supplementation.

Thank you very much for your comments. We deleted the sentence including the word “hypoxia”.

Line 82 Hypothermia induced by 2g load is due to decreased plasma catecholamine level “ but no info about catecholamine in the abstract.

Thank you very much for your comments. We added the information of plasma catecholamine in the abstract.

Lines 83-88 This part is rather introduction but not the results

Thank you very much for your comments. We changed the text as a reviewer suggested (line 105-107).

Also, major concern is that all changes in 2g load-induced hypothermia can be a result of hypoxia. As it was shown that hypergravity loading was accompanied with hypoxia (Borisova and Krisanova 2008, *Advances in Space Research* 42).

Thank you very much for your constructive comments. We did the additional experiments to examine whether the hypoxia was occurred during 2g loading. We inserted the catheter into the common carotid artery in 4 mice then we did the blood sampling via the swivel (Instech Laboratories, Inc.) during 2g load. The blood sampling was conducted 30 min after 2g loading start. The blood was transferred to the blood gas analyzer (IDEXX Laboratories) to measure the partial pressure of both oxygen (PO₂) and carbon dioxide (PCO₂) and oxygen saturation of blood (SO₂). All values were not changed by 2g load (please see the figure below), suggesting that hypoxia was not occurred during 2g load. The citation which a reviewer indicated used 10g instead of 2g, thus the magnitude of the gravity might be affect the condition of the blood.

206 “activity in the 2g environment was significantly higher in VGLUT2-expressing neurons-deleted mice than in the control group” and this fact can be explained by a decrease in amount of excitotoxic glutamate during 2g-induced hypoxia.

Thank you very much for your comments. Hypoxia was not occurred in the present study.

217-219 “Accordingly, we hypothesized that use of chemogenetic stimulation of VGLUT2-expressing neurons present in the VNC is a useful tool for training and that the

hypothermic response may consequently be attenuated.” But it has been suggested that hypothermia is a useful compensatory mechanism.

Thank you very much for your comments. We are sorry for your confusing. We revised the text (line 240).

230-231 “Exposure to the 2g for 9 h, which was applied 2 days prior to the start of the experiment, significantly attenuated the 2g- induced hypothermia” .This fact can also indicate hypoxia preconditioning training.

Thank you very much for your comments. Hypoxia was not occurred during 2 g load.

Reviewers' comments:

Reviewer #1 (Remarks to the Author):

The authors have addressed my concerns in full, other than 2 relatively minor issues.

1. The CNO dose used (3.0mg/kg) is quite high. Did the authors run CNO controls? If so (and even if not shown) this should be described in the text.

2. The authors added a sentence on 114-115 to recognize the clear absence of effect of +G on the circadian rhythm of body temperature in the vestibular lesion group. I understand that rhythms (or analysis thereof) were not a focus of this study, but this is a really interesting observation and a nice confirmation of prior work. I appreciate the addition of the sentence and would suggest the following slight modification, "The peripheral vestibular organs may also have a potential influence on the regulation in BT rhythms (Figs. 1a and 1b), which is consistent with prior findings (Fuller and Fuller, 2006)."

Reviewer #2 (Remarks to the Author):

The paper and worked were properly improved.

The additional experiments are welcome.

I have no more suggestions or question.

Reviewer #3 (Remarks to the Author):

The MS "Role of glutamatergic neurons in the vestibular nuclear complex on the gravitational stress-induced hypothermia in mice" was restructured and now it looks focused.

The research topic is original and can be interesting for a wide cohort of scientists not only in space-related area but also for those involved in therapeutic hypothermia treatment.

Despite the potential importance and broad auditory of readers, the MS in the current form cannot be published and needs major revision.

Major concerns

The authors performed additional experiments and responded to the comment:

Comment:

Also, major concern is that all changes in 2g load-induced hypothermia can be a result of hypoxia. As it was shown that hypergravity loading was accompanied with hypoxia (Borisova and Krisanova 2008, Advances in Space Research 42).

Response: Thank you very much for your constructive comments. We did the additional experiments to examine whether the hypoxia was occurred during 2g loading. We inserted the catheter into the common carotid artery in 4 mice then we did the blood sampling via the swivel (Instech Laboratories, Inc.) during 2g load. The blood sampling was conducted 30 min after 2g loading start. The blood was transferred to the blood gas analyzer (IDEXX Laboratories) to measure the partial pressure of both oxygen (PO₂) and carbon dioxide (PCO₂) and oxygen saturation of blood (SO₂). All values were not changed by 2g load (please see the figure below), suggesting that hypoxia was not occurred during 2g load. The citation which a reviewer indicated used 10g instead of 2g, thus the magnitude of the gravity might be affect the condition of the blood.

New comment:

The explanation is not completely satisfactory. Hypergravity-induced hypoxia can be some type of circulatory hypoxia, when the lungs are working just fine and the blood can carry sufficient oxygen, but due to changes in circulations, the brain has not sufficient amount of blood.

Minor concerns

Line 41 "Vesicular glutamate transporters" – Why it is from capital letter?

Line 455 – do not use "we showed..", rather "it was shown.." in the MS text.

Summary

It is absolutely unclear statement:

436 Hypothermia is thought to be a result natural selection. Chronic activation of VGLUT2-expressing neurons in the VNC 437 and C1 neurons may be associated with this selection.

Reviewers' comments:

Reviewer #1 (Remarks to the Author):

The authors have addressed my concerns in full, other than 2 relatively minor issues.

1. The CNO dose used (3.0mg/kg) is quite high. Did the authors run CNO controls? If so (and even if not shown) this should be described in the text.

Thank you very much for your comment. We have already confirmed the effect of CNO (3 mg/kg) injection on the movement and body temperature in C57BL/6J mice (n = 3). No effects were observed. This information is added in the text (line 931-932).

2. The authors added a sentence on 114-115 to recognize the clear absence of effect of +G on the circadian rhythm of body temperature in the vestibular lesion group. I understand that rhythms (or analysis thereof) were not a focus of this study, but this is a really interesting observation and a nice confirmation of prior work. I appreciate the addition of the sentence and would suggest the following slight modification, “The peripheral vestibular organs may also have a potential influence on the regulation in BT rhythms (Figs. 1a and 1b), **which is consistent with prior findings (Fuller and Fuller, 2006).**”

Thank you very much for your comments. The sentence was added in the text (line 114-115).

Reviewer #2 (Remarks to the Author):

The paper and worked were properly improved.

The additional experiments are welcome.

I have no more suggestions or question.

I would like to thank your constructive comments so far to improve the manuscript.

Reviewer #3 (Remarks to the Author):

The MS “Role of glutamatergic neurons in the vestibular nuclear complex on the gravitational stress-induced hypothermia in mice” was restructured and now it looks focused. The research topic is original and can be interesting for a wide cohort of scientists not only in space-related area but also for those involved in therapeutic hypothermia treatment. Despite the potential importance and broad auditory of readers, the MS in the current form cannot be published and needs major revision.

Major concerns

The authors performed additional experiments and responded to the comment:

Comment:

Also, major concern is that all changes in 2g load-induced hypothermia can be a result of hypoxia. As it was shown that hypergravity loading was accompanied with hypoxia (Borisova and Krisanova 2008, *Advances in Space Research* 42).

Response: Thank you very much for your constructive comments. We did the additional experiments to examine whether the hypoxia was occurred during 2g loading. We inserted the catheter into the common carotid artery in 4 mice then we did the blood sampling via the swivel (Instech Laboratories, Inc.) during 2g load. The blood sampling was conducted 30 min after 2g loading start. The blood was transferred to the blood gas analyzer (IDEXX Laboratories) to measure the partial pressure of both oxygen (PO₂) and carbon dioxide (PCO₂) and oxygen saturation of blood (SO₂). All values were not changed by 2g load (please see the figure below), suggesting that hypoxia was not occurred during 2g load. The citation which a reviewer indicated used 10g instead of 2 g, thus the magnitude of the gravity might be affect the condition of the blood.

New comment:

The explanation is not completely satisfactory. Hypergravity-induced hypoxia can be some type of circulatory hypoxia, when the lungs are working just fine and the blood can carry sufficient oxygen, but due to changes in circulations, the brain has not sufficient amount of blood.

Thank you very much for your comment. Our previous work showed that the cerebral blood flow was maintained in both hypergravity (1.1~1.2g) and microgravity using parabolic flights in rats (Tanaka et al., *J Appl Physiol*, 2005). The other experiment showed that slight increase of cerebral cortical blood flow in rabbits was observed in hypergravity (1.8-2.0g) using parabolic flights (Florence et al., *Eur J Appl Physiol Occup Physiol*. 1998). The effect of changes in hydrostatic pressure gradient induced by hypergravity might be lower in rodents and rabbits than human. Accordingly, it is possible that the amount of blood in the brain is sufficient in 2g environment which we used in the present study.

Minor concerns

Line 41 “Vesicular glutamate transporters” – Why it is from capital letter?

Thank you very much for your comment. I revised the word (line 41)

Line 455 – do not use “we showed..”, rather ” it was shown..” in the MS text.

Thank you very much for your comment. The indicated phrase was not found in the text (including line 455).

Summary

It is absolutely unclear statement:

436 Hypothermia is thought to be a result natural selection. Chronic activation of VGLUT2-expressing neurons in the VNC 437 and C1 neurons may be associated with this selection.

Thank you very much for your comment. I deleted the sentence (line 436).

REVIEWERS' COMMENTS:

Reviewer #3 (Remarks to the Author):

All comments were addressed. The MS can be accepted.